# EVALUATING REPRESENTATIONS WITH READOUT MODEL SWITCHING

**Yazhe Li**
yazhe@deepmind.com

**Jorg Bornschein**
bornschein@deepmind.com

**Marcus Hutter**
mhutter@deepmind.com

## ABSTRACT

Although much of the success of Deep Learning builds on learning good representations, a rigorous method to evaluate their quality is lacking. In this paper, we treat the evaluation of representations as a model selection problem and propose to use the Minimum Description Length (MDL) principle to devise an evaluation metric. Contrary to the established practice of limiting the capacity of the readout model, we design a hybrid discrete and continuous-valued model space for the readout models and employ a switching strategy to combine their predictions. The MDL score takes model complexity, as well as data efficiency into account. As a result, the most appropriate model for the specific task and representation will be chosen, making it a unified measure for comparison. The proposed metric can be efficiently computed with an online method and we present results for pre-trained vision encoders of various architectures (ResNet and ViT) and objective functions (supervised and self-supervised) on a range of downstream tasks. We compare our methods with accuracy-based approaches and show that the latter are inconsistent when multiple readout models are used. Finally, we discuss important properties revealed by our evaluations such as model scaling, preferred readout model, and data efficiency.

## 1 INTRODUCTION

Data representation is crucial to the performance of machine learning algorithms (Bengio et al., 2013). Much of the success of Deep Neural Networks (DNN) can be attributed to their capability of gradually building up more and more abstract representations (Lee et al., 2009). In supervised learning, although the network is trained to predict a specific aspect of the input, the intermediate representations are often proven to be useful for many other downstream tasks (Yosinski et al., 2014). In unsupervised and self-supervised learning, the network is trained on a surrogate task, such as reconstruction (Hinton & Salakhutdinov, 2006; Kingma & Welling, 2013; He et al., 2021) and contrastive prediction (van den Oord et al., 2018; Chen et al., 2020), which is supposed to capture generic prior of the data. In recent years, there has been significant improvements in unsupervised representation learning with state-of-the-art models achieving performance comparable to its supervised counterpart (Tomasev et al., 2022).

Despite the importance of data representation, the evaluation method for representations is rarely discussed. The most prevalent practice is to train a readout model on the downstream task. The readout model often has a shallow architecture, e.g. linear layer, to limit its capacity, so that the task performance reflects the representation quality. The problem with this approach is that the readout model cannot adapt to the nature of the representations. Deeper models and fine-tuning alleviate this issue. However, the representations are left with multiple metrics, each using a different readout mechanism, making the comparison extremely difficult (Nozawa & Sato, 2022).

In this paper, we treat evaluating representations as a model selection problem. We propose to use *Minimum Description Length* (MDL) as the main evaluation metric and use model switching to accommodate the need for multiple readout models. MDL is a well-studied compression-based approach for inductive inference that provides a generic solution to the model selection problem (Rissanen, 1984; Grunwald, 2004; Wallace, 2005; Solomonoff, 1964; Rathmanner & Hutter, 2011). MDL performs a similar role as held-out validation does for *Emperical Risk Minimization* (Vapnik, 1991), but has the advantage of being able to deal with single sequence and non-stationary data.

It is closely related to Bayesian model selection and includes a form of Occam's Razor where the metric takes into account the model complexity. The complexity term can be explicitly represented as the codelength of the model in the case of a 2-part code, as a KL-term when using a variational code, or implicitly when using prequential or Bayesian codes. By including the model complexity in the evaluation metric, we automatically resolve the need of limiting the readout model complexity and are able to compare MDL scores freely between different readout mechanisms. Intuitively, if the induced representation is nonlinear and requires a higher capacity model for readout, the MDL score reflects this by having a larger complexity term. Note that this also applies in the case of fine-tuning, where the pre-trained model is allowed to adapt for the downstream tasks. Model switching allows multiple readout models and automatically finds the best readout model for the downstream task at each dataset size (Figure 1). Therefore, MDL with readout model switching provides a unified framework for evaluating representations regardless the evaluation protocol employed.

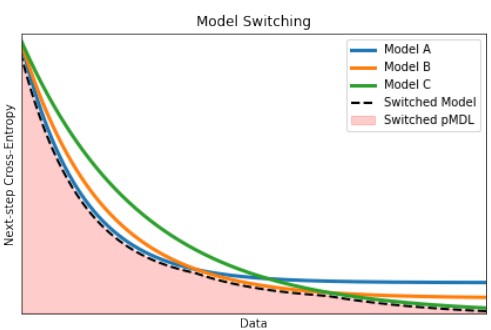

Figure 1: Illustration of switching between models of different complexity: Depending on the number of training examples either $A$, $B$, or $C$ has the best generalization performance. An optimally switched model will have the best performance at each point and thus the lowest prequential description length (= area under the curve).

It is conjured that useful representations make the variability in the data more predictable and allow to efficient learning human-like data (Hénaff et al., 2019). The MDL evaluation metric formalizes the data efficiency perspective - especially evident in the form of prequential MDL. Prequential MDL (Dawid & Vovk, 1999; Poland & Hutter, 2005) turns computing the description length $L(\mathcal{D}|\phi) = -\log p(\mathcal{D}|\phi)$ into a sequential prediction problem: $\log p(\mathcal{D}|\phi) = \sum_t \log p(y_t|\phi_{\leq t}, y_{<t})$, where $\phi$ is an encoder for feature extraction, $\phi_t := \phi(x_t)$ an encoded input, and $(x_{<t}, y_{<t})$ is the data seen before timestep $t$. In order to achieve a short description length, it is beneficial if the representation is capable of fast learning, i.e. given a good representation, a few examples are enough to achieve good performance. As a by-product of our readout model switching based on MDL, we can visualize the predictive performance along the data sequence and inspect how efficient the downstream learning is.

Our contributions are as follows:

1. We propose readout model switching for evaluating representations.
2. We prove a regret bound of readout model switching and discuss assumptions made.
3. We use an online learning framework for efficient computation and evaluate the performance of several popular visual representation methods on the set of downstream tasks.
4. We investigate and inspect the preferred readout models, data efficiency and model scaling.

## 2 BACKGROUND

**Minimum Description Length** is based on the fundamental idea that learning and comprehension correspond to compression (Rathmanner & Hutter, 2011). Given data $\mathcal{D}=(y_t)_1^N \in \mathcal{Y}^N$ and a hypothesis space $\mathcal{M} = \{M_1, M_2, \dots\}$, where each hypothesis $M$ corresponds to a parametric probabilistic model $p(\mathcal{D}|\theta, M)$, MDL aims to identify the model that can compress the data $\mathcal{D}$ best. Considering the close relationship between lossless coding and probability distributions, this can be achieved by associating a codelength function $L(\mathcal{D}|M)=-\log p(\mathcal{D}|M)$ with each hypothesis. *A vast body of literature shows that models with a shorter description length have a better chance of generalizing to future data* (Wallace, 2005; Grünwald, 2007a; Rathmanner & Hutter, 2011).

A crude way to obtain description lengths is to consider $L(\mathcal{D}\,|\,M)=L_M(\theta)+L_M(\mathcal{D}\,|\,\theta)$, where $L_M(\theta)$ is the cost of encoding the parameters and $L_M(\mathcal{D}|\theta)=-\log p(\mathcal{D}|\theta, M)$ is the cost of compressing the data with the parameterized model. This *two-part code* approach is intuitive but suboptimal and ambiguous because it does not specify how to encode the parameters. This crude MDL approach has been refined in three distinct but closely related ways:

1) The Bayesian marginal likelihood:
   $L_{\text{Bayes}}(\mathcal{D}|M) := -\log \int_\theta p(\mathcal{D}|\theta, M)p(\theta)d\theta$ and its variational upper bound $L_{\text{ELBO}}(\mathcal{D}|M) :=$ $\text{KL}(q(\theta)|p(\theta)) - \mathbb{E}_{\theta \sim q}\log p(\mathcal{D}|\theta, M)$. Both depend on the chosen prior $p(\theta)$ and require access to the posterior distribution over parameters given data; or an approximation thereof.

2) Normalized Maximum Likelihood (NML):
   $L_{\text{NML}}(\mathcal{D}|M) := -\log \left[ p(\mathcal{D}|\hat{\theta}_{\text{ML}}(\mathcal{D}), M) \,/\, \int_z p(\mathcal{D}|\hat{\theta}_{\text{ML}}(z), M)dz \right]$ only relies on the maximum likelihood estimator $\hat{\theta}_{\text{ML}}(\cdot)$, but normalizes over all potentially observable data $z \in \mathcal{Y}^N$, which is often intractable and even undefined for many model families of interest. The log-numerator is also called the complexity of $M$ and measures how well the model could fit all possible data. Under NML, a model $M$ achieves a short description length only when it fits the given data well (high $\log p(\mathcal{D}|\hat{\theta}_{\text{ML}}(\mathcal{D}), M)$) as well as not fit well many different data (low denominator).

3) The prequential approach:
   $L_{\text{plugin}}(\mathcal{D}|M) := -\sum_{t=1}^{N} \log p(y_t|\hat{\theta}(\mathcal{D}_{<t}), M)$ decomposes the description length over datapoints and relies on the choice of a suitable plug-in parameter estimator $\hat{\theta}(\mathcal{D}_{<t})$. It emphasizes that in order to achieve a short codelength, a model $M$ must not only be a good predictor given all training data $\mathcal{D}$, but already given only parts of the data $\mathcal{D}_{<t}$ for all $t$, i.e. be a sample efficient predictor. Model complexity and sample efficiency can thus be considered two sides of the same coin.

Approaches to computing description lengths for some data $\mathcal{D}$ under a model family $p(\mathcal{D}|\theta, M)$ are called *universal codes* when they result in description lengths that stay close to the (generally unachievable) maximum likelihood codelength for that family: $L_{\text{universal}}(\mathcal{D}|M) \leq -\log p(\mathcal{D}|\hat{\theta}_{\text{ML}}(\mathcal{D}), M) + \mathcal{O}(\log N)$. $L_{\text{NML}}(\mathcal{D}|M)$ is by definition universal; $L_{\text{Bayes}}(\mathcal{D}|M)$ and $L_{\text{plugin}}(\mathcal{D}|M)$ are universal for many reasonable choices of the prior $p(\theta)$ and the estimator $\hat{\theta}(\mathcal{D}_{<t})$ respectively (Grünwald, 2007b).

**Methods for Representation Evaluation.** Linear probing, using a linear layer for readout, is the most prevalent evaluation protocol for representations, the performance of which is reported and compared in almost all of the recent representation learning works (van den Oord et al., 2018; Chen et al., 2020; Grill et al., 2020; Li et al., 2021; Caron et al., 2021; He et al., 2021). Apart from linear probing, clustering algorithms such as Hebbian linear classifier (Bengio, 2012) and K-Nearest Neighbor (Caron et al., 2021) are also popular choices. However, concerns have been raised for model comparison using these simple probes. Resnick et al. (2019) shows that the performance of linear probing can be misleading, since models performing weakly with linear probing can perform better under alternative protocols. He et al. (2021) also argue that the use of linear probes limits the development of methods that induce non-linear representations. Hewitt & Liang (2019) question the practice of using downstream accuracy as the evaluation metric and propose to design probes based on their selectivity, which puts the task accuracy in context with the probe's capacity to memorize. Yogatama et al. (2019) and Voita & Titov (2020) propose to use MDL, either variational or prequential code, as the evaluation metric, which takes into account the complexity in the final measurement. Hénaff et al. (2019) bring to light the benefit of pre-training for downstream data efficiency. Whitney et al. (2020) propose a modified version of MDL that introduces a baseline corresponding to solving the downstream task. While previous works proposing MDL as an evaluation metric use only a single readout protocol, our method can be applied for combining different readout protocols regardless of the backbone being frozen or fine-tuned. Our use of an online framework to compute prequential MDL scores which has not been used previously to evaluate representations leads to shorter description lengths (Bornschein et al., 2022) and crucially enables the model switching between readout-methods at each timestep. As a result, our framework gives insights into properties of the representations that can guide us in choosing the best way to use the representation.

## 3   EVALUATING REPRESENTATIONS WITH READOUT MODEL SWITCHING

We formally define the model selection problem for representation evaluation under the MDL principle as follows: the objective is to compare the encoder $\phi(.)$ that minimizes the codelength $L(\mathcal{D}|\phi) = -\log p(\mathcal{D}|\phi)$, where $\mathcal{D} = \{(x_t, y_t)\}_1^N$ is the inputs $x_t$ and associated prediction targets $y_t$. Prequential MDL decomposes the codelength into the cumulative prediction loss

$L(\mathcal{D}|\phi) = -\sum_{t=1}^{N} \log p(y_t|\phi_{\leq t}, y_{<t})$, where $\phi_t := \phi(x_t)$ is the encoded feature of input $x_t$. In order to perform the prediction task from the features, previous works have to fix the probe to be a linear layer or a shallow MLP. In our study, instead of picking a fixed model, we design a hybrid of a continuous and discrete-valued $(k, \theta_k)$ model space of the readout models, where "expert" $k \in [1, \ldots, K]$ is the model class and $\theta_k$ are the continuous-valued parameters corresponding to the model class $k$. Each readout model can be seen as an expert forecasting system that learns to predict the data independently in an online fashion. At each datapoint $x_t$, the prediction $\hat{y}_t$ is a combination of the experts' predictions. The final MDL score is the cumulative predictive performance of the board of $K$ experts.

By constructing the probe using multiple readout models and switching between them, the final MDL score is marginalized over the choice of the readout architectures. Since MDL automatically takes into account the complexity of the model, i.e. any extra complexity used is going to cost in the MDL score, our framework allows the representation to select the most suitable readout mechanism at any data size while still maintaining a comparable final score for the representations. In addition, after computing the score, we can inspect the posterior over the readout models at any given data size, as well as data efficiency along the trajectory, providing valuable insight into the characteristics of the representation.

**Readout Model Switching.** Given $K$ readout models, we define random variables $\xi_t \in [1, \ldots, K], \forall t \in [1, \ldots, N]$ the readout model to use at timestep $t$. The Expert Sequence (ES) prior (Koolen & de Rooij, 2008) uses a hidden Markov model (HMM) to define the joint distribution of the data $y_{1:N}$ and random variables $\xi_{1:N}$ as $p(y_{1:N}, \xi_{1:N}|\phi) = \prod_{t=1}^{N} p_{\xi_t}(y_t|\phi_{\leq t}, y_{<t})p(\xi_t|\xi_{t-1})$, where $p(\xi_1|\xi_0)=p(\xi_1)$ is the initial distribution and $p_k(\cdot)$ is the prediction of the $k$th readout model. At each timestep $t$, we use the previously observed data to estimate $\hat{\theta}_k(\phi_{<t}, y_{<t})$ for the parameters of the readout models. Therefore, the $k$th readout model prediction $p_k(y_t|\phi_{\leq t}, y_{<t})$ is the plugin distribution $p(y_t|\phi_t, \hat{\theta}_k(\phi_{<t}, y_{<t}))$. The final readout model switching codelength function has the form:

$$L_{\text{Switch}}(y_{1:N}|\phi) = -\log p(y_{1:N}|\phi) = -\log \sum_{\xi_{1:N}} \prod_{t=1}^{N} p(y_t|\phi_t, \hat{\theta}_{\xi_t}(\phi_{<t}, y_{<t}))p(\xi_t|\xi_{t-1}) \quad (1)$$

The readout model switching defined in Equation 1 represents a family of codes that combines prequential and Bayesian MDL. From the perspective of the HMM, $p(\xi_t|\xi_{t-1})$ is the transition matrix and, in our context, it is equivalent to a switching strategy between the readout models. Every switching strategy corresponds to a specific code in the family. For example, if $p(\xi_t|\xi_{t-1}) = 1$ iff $\xi_t = \xi_{t-1}$ (zero otherwise), i.e when we believe a single readout model is responsible for modeling the whole data sequence, then Equation 1 simplifies to a Bayesian mixture code $L_{BM}(y_{1:N}|\phi) = -\log \sum_k p(\xi_1 = k) \prod_{t=1}^{N} p(y_t|\phi_t, \hat{\theta}_k(\phi_{<t}, y_{<t}))$.

For the rest of the paper, we are going to use Fixed Share (FS) (Herbster & Warmuth, 1998; 2004) as the switching strategy. The prior defined in fixed share can be interpreted as follows: at timestep $t$, with a probability $1 - \alpha_t$, the expert is the same as in the previous timestep; with probability $\alpha_t$, it switches to another model according to the probability $w(k)$. We are going to use the uniform distribution $w(k) = 1/K$ through out the paper. The switching strategy can be simplified and expressed as

$$p(\xi_t|\xi_{t-1}) = \begin{cases} 1 - \frac{K-1}{K}\alpha_t & \text{if } \xi_t = \xi_{t-1} \\ \frac{1}{K}\alpha_t & \text{if } \xi_t \neq \xi_{t-1} \end{cases} \quad \text{where} \quad \alpha_t := \frac{m-1}{t} \in [0, 1] \quad (2)$$

is decreasing in timestep $t$, and $m$ is a hyperparameter. Note that there are other switching strategies that could be used, e.g. elementwise mixture, fixed share with constant switching rate, switch distribution and run-length model (Koolen & de Rooij, 2008). We provide a comparison of these strategies in Appendix A.6. In general, we find that Bayesian mixture and elementwise mixture perform significantly worse than fixed share and other more sophisticated switching strategies. The poor codelength of the Bayesian mixture is due to the catch-up phenomena (van Erven et al., 2012): it assumes that a single model is responsible for modeling the whole data sequence, while evidently the best model changes with data set size.

**Regret Bound.** To characterize the gap between the codelength for our readout model switching and the shortest codelength that can be achieved in hindsight, we derive a regret bound in Theorem 1.

This bound tells us whether the readout model switching converges to the optimal solution and, if so, how fast it converges.

**Theorem 1.** *Let a sequence $x^N = (x_i)_{i=1}^N$ be sampled i.i.d. from distribution $P$. Let all readout models $M_1, \ldots M_K$ be in the exponential family. Let $\hat{\theta}_k(x^t)$ be the regularized Maximum Likelihood (ML) estimator of $M_k$ after the first $t$ examples $x^t = (x_i)_{i=1}^t$. The expected regret of readout model switching w.r.t. the best sequence of readout models $\xi_{1:N}^*$ and their best fit parameters $\hat{\theta}_{\xi_t^*}(x^N)$ is*

$$R(N) := \mathbb{E}_P \left[ \sum_{t=1}^N \log M_{\xi_t^*}(x_t; \hat{\theta}_{\xi_t^*}(x^N)) - \log \sum_{\xi_{1:N}} p(\xi_{1:N}) \prod_{t=1}^N M_{\xi_t}(x_t | \hat{\theta}_{\xi_t}(x^{t-1})) \right]$$

$$\leq -\mathbb{E}_P \left[ \log p(\xi_{1:N}^*) \right] + \frac{1}{2} \max_k \left\{ Tr \left( Cov_{M_{\mu_k^*}}^{-1}[T_k(x)] Cov_P [T_k(x)] \right) \right\} \log N \qquad (3)$$

*where $\mu_k^* = \mathbb{E}_P[T_k(x)]$, and $T_k(.)$ is the sufficient statistic and $\mu_k$ is the mean value parameter of $M_k$.*

The proof of Theorem 1 can be found in Appendix A.1. The expected regret consists of two terms: the regret of the switching strategy and the regret from the plugin distribution. The latter is bounded by $C \log N$ with $C$ being a constant determined by the model family and the true distribution. For Fixed Share with decreasing switching rate, $-\mathbb{E}_P \left[ \log p(\xi_{1:N}^*) \right] \leq m \log K + (m-1) \log N/(m-1)$, where $m - 1$ is the number of switch points (Veness et al., 2012; Koolen & de Rooij, 2013). Therefore, the expected regret is of the order of $\log N = o(N)$, making the corresponding code a universal code. Many other switching strategies have the same property, such as Bayesian mixture and run-length code (Koolen & de Rooij, 2013).

Like many theories and algorithms applied to DNN, some of the assumptions of Theorem 1 do not hold in practice: DNNs are not in the exponential family and the solutions found by Stochastic Gradient Descent (SGD) training are not ML solutions. Although the exact nature of the final codelength heavily relies on the training algorithm and much of the understanding of SGD is left for future works, as long as the estimator converges asymptotically to the optimal solution, we can use the code for model selection. In fact, there exist estimators that outperform the ML estimator such that the resulting prequential code better approximates the Normalized Maximum Likelihood (NML) code which is the optimal universal code (Grünwald & Kotlowski, 2010).

**Computing MDL with Online Learning.** Algorithm 1 describes the online implementation for computing the description length and for performing inference over the preferred readout models $p(\xi_t | \mathcal{D}_{<t})$. For each example in the sequence we first obtain a marginal prediction by weighting the predictions from the readout models according to our current belief over the mixture. After

---

**Algorithm 1** MDL with Readout Model Switching

---

**Require:** data $\mathcal{D} = (x_t, y_t)_{t=1}^N$; $K$ readout models; initial model probability $p(\xi_1)$ and switching strategy $p(\xi_t | \xi_{t-1})$ that updates the parameters given a dataset *UpdateParameters*
1: Initialize: model parameters $\theta_1, \ldots, \theta_K$; $s = \log p(\xi_1)$; an empty list $Q$ to store posterior $p(\xi_t | \mathcal{D}_{<t})$
2: **for** $t = 1$ **to** $N$ **do**
3:     Compute $\log p(\xi_t, y^{t-1} | x^{t-1})$: $s \leftarrow \log \sum_{\xi_{t-1}} \exp \left( \log p(\xi_t | \xi_{t-1}) + s \right)$
4:     Compute posterior $p(\xi_t | \mathcal{D}_{<t})$ and store in $Q$:
    $\log p(\xi_t | \mathcal{D}_{<t}) = \log p(\xi_t, y^{t-1} | x^{t-1}) - \log \sum_{\xi_t} p(\xi_t, y^{t-1} | x^{t-1})$
5:     Compute next-step loss of $K$ readout models: $\mathcal{L}_t^k := -\log p_k(y_t | x_t) = -\log p(y_t | x_t; \theta_k)$
6:     Combine $K$ models to update $\log p(\xi_t, y^t | x^t)$: $s \leftarrow -\mathcal{L}_t^{\xi_t} + s$
7:     **for** $k = 1$ **to** $K$ **do**
8:         Update parameters: $\theta_k \leftarrow UpdateParameters(\theta_k, \mathcal{D}_{\leq t})$
9:     **end for**
10: **end for**
11: Compute total codelength $\mathcal{L}_{\text{switch}} \leftarrow -\log \sum_{\xi_N} \exp(s)$
12: **return** $\mathcal{L}_{\text{switch}}$ and $Q$

---

receiving the label $y_t$, we compute the cross-entropy loss for the marginal prediction and add it to the cumulative loss. Then, the belief over the mixture is updated and the new example is added to the training data for readout model parameter estimation. The latter is done by online SGD with replay.

For the posterior belief over the model switches $p(\xi_t|\mathcal{D}_{<t})$ the above procedure amounts to the well known, efficient, and exact forward pass HMM inference algorithm. For the neural network parameters $\theta_k$ on the other hand we use SGD as the plugin estimator for prequential MDL. Prior work has shown that SGD is an effective and reliable plug-in estimator for modern neural networks (Blier & Ollivier, 2018). Note that most prior works on computing prequential MDL scores with neural networks use a block-wise approximation; i.e. they partition the data into a small number of segments and evaluate each segment independently by training on all the previous segments to convergence (Voita & Titov, 2020; Bornschein et al., 2020; Whitney et al., 2020). Since we need a prediction at each step $t$ to update the mixture weights, we here instead use the online mini-batch incremental training with replay described in Bornschein et al. (2022). This approach updates the parameters continually with online mini-batch gradient descent training on the new and previously seen examples. Instead of using a replay buffer to store examples in-memory, we use replay streams to iterate through the examples in their original order from permanent storage. By tuning the number of streams and the reset probability, we can adjust the amount of replay training and change from uniform sampling to a sampling strategy that is biased towards more recent samples. However, in this work, we use uniform random replay samples for all our experiments. As suggested in Bornschein et al. (2022), we use Exponential Moving Average (EMA) and label smoothing to improve performance of the online training, but not weight standardization as we don't want to alter the architectures.

A strictly equivalent alternative to the fully online implementation is a two-stage implementation (see Appendix A.3): First, the prequential online parameter estimation is computed for each of the readout models independently while the per-step predictions and log-losses are recorded for later analysis. In a second stage, the predictions and losses can be used to compute the mixtures, switching posterior $p(\xi_t|\mathcal{D}_{<t})$ and overall description lengths. To facilitate experimenting with different switching strategies we use this 2-stage implementation for all our experiments.

## 4 EXPERIMENTS

First, we compare our evaluation framework with linear and a combination of MLP evaluations using vision encoders pre-trained on ImageNet (Deng et al., 2009) with different objectives. For this purpose, we use the VTAB benchmark (Zhai et al., 2019) as downstream tasks. Then, we use ImageNet classification as downstream task to showcase the insights, such as data efficiency and preferred readout model, and for comparing in-domain and out-of-domain transfer, revealed by our evaluation method.

The VTAB benchmark consists of 19 datasets that can be grouped into three categories: natural, specialized and structured. For some VTAB datasets, notably KittiDistance, SmallNORBAzimuth, PatchCamelyon, we discover that there is a distribution shift between train and test subsets (see details in Appendix A.2). Therefore, in order to make sure that data are truly i.i.d, we reshuffle the train and test data and use the reshuffled datasets for all our experiments.

To summarize the performance over multiple datasets, we follow the procedure recommended by Demšar (2006) and report the average rank of the representations: for each dataset, we rank the representations by the evaluation metric (accuracy or MDL score); then, we compute the average rank of a representation over the 19 datasets. For any pair of representations $A$ and $B$, the difference of the average rank can serve as a statistic for a hypothesis test with H0: $A$ performs the same as $B$. If the hypothesis is rejected, we conclude the representations perform significantly different. We adopt the Nemenyi test (Nemenyi, 1963) which suggests that the null-hypothesis is rejected if the average ranks differ by at least a critical value. This critical value is computed as $q_\gamma\sqrt{r(r+1)/6N}$, where $q_\gamma$ is a constant depending on the confidence level $\gamma$, $r$ is the number of representations to compare and $N$ is the number of datasets. In the following comparisons, we also report the critical value with a confidence level of $\gamma = 0.1$.

We use the online learning framework as described in Section 3 and compute the Fixed Share code length with decreasing switching rate $\alpha_t$ defined in (2). The raw MDL scores per dataset are reported in Appendix A.7. For ImageNet, we use readout model switching between 4 model architectures: Linear and 1-3 layer MLPs. For VTAB, we include up to 7-layer MLP's as readout architectures. We furthermore instantiate each of these architectures with all hyperparameter settings from the hyperparameter grid outlined in Appendix A.5. The model switching is therefore performed between $K = \#\text{architectures} \times \#\text{hyperparameters}$ many independent models. We report results using a switching rate $m = 2$ for ImageNet and $m = 11$ for VTAB. The switching rates were chosen to optimize the overall description length.

**Pre-training Objectives.** We compare downstream performance of different pre-training objectives on the VTAB benchmark. To make the comparison, we choose two backbone architectures, ResNet-50 v1 (He et al., 2015) and ViT/B16 (Dosovitskiy et al., 2020), and only change the pre-training objectives. For ResNet-50, we pre-train on supervised SimCLR and BYOL losses; for ViT/B16, we pre-train on supervised (Touvron et al., 2020; He et al., 2021) DINO and MAE losses. SimCLR (Chen et al., 2020) is a self-supervised method based on contrastive learning. BYOL (Grill et al., 2020) and DINO (Caron et al., 2021) both use a teacher network. BYOL is a negative-free method for self-supervised representation, while DINO uses the ViT architecture and learns through self-distillation. MAE (He et al., 2021) uses masked reconstruction tasks to learn representation.

<table>
<tr><td colspan="8" align="center">(a) Pre-training Objectives</td></tr>
<tr><td>Architecture</td><td>Objective</td><td>Linear</td><td>#</td><td>MLP</td><td>#</td><td>MDL</td><td>#</td></tr>
<tr><td>ViT/B16</td><td>DINO</td><td>2.26</td><td>1</td><td>1.84</td><td>1</td><td>1.89</td><td>1</td></tr>
<tr><td>ResNet-50</td><td>BYOL</td><td>2.37</td><td>2</td><td>2.63</td><td>2</td><td>2.53</td><td>2</td></tr>
<tr><td>ViT/B16</td><td>MAE</td><td>3.84</td><td>4</td><td>3.24</td><td>3</td><td>3.79</td><td>3</td></tr>
<tr><td>ViT/B16</td><td>SUP</td><td>4.18</td><td>5</td><td>4.21</td><td>4</td><td>4.16</td><td>4</td></tr>
<tr><td>ResNet-50</td><td>SimCLR</td><td>4.53</td><td>6</td><td>4.63</td><td>6</td><td>4.21</td><td>5</td></tr>
<tr><td>ResNet-50</td><td>SUP</td><td>3.82</td><td>3</td><td>4.45</td><td>5</td><td>4.42</td><td>6</td></tr>
</table>

ResNet and ViT with different pre-training objective and model sizes. The threshold of the rank difference to be significant for hypothesis testing is 1.894 (left) and 3.65 (right).

<table>
<tr><td colspan="8" align="center">(b) Model Sizes</td></tr>
<tr><td>Architecture</td><td>Obj.</td><td>Linear</td><td>#</td><td>MLP</td><td>#</td><td>MDL</td><td>#</td></tr>
<tr><td>ResNet-101</td><td>BYOL</td><td>2.84</td><td>1</td><td>3.92</td><td>3</td><td>3.05</td><td>1</td></tr>
<tr><td>ResNet-152</td><td>BYOL</td><td>3.11</td><td>2</td><td>3.71</td><td>2</td><td>3.26</td><td>2</td></tr>
<tr><td>ViT/L16</td><td>MAE</td><td>3.42</td><td>3</td><td>2.53</td><td>1</td><td>3.74</td><td>3</td></tr>
<tr><td>ResNet-50</td><td>BYOL</td><td>4.24</td><td>4</td><td>4.26</td><td>4</td><td>4.05</td><td>4</td></tr>
<tr><td>ViT/B16</td><td>SUP</td><td>7.34</td><td>9</td><td>6.47</td><td>6</td><td>6.00</td><td>5</td></tr>
<tr><td>ViT/B16</td><td>MAE</td><td>6.89</td><td>5</td><td>5.34</td><td>5</td><td>6.21</td><td>6</td></tr>
<tr><td>ResNet-50</td><td>SUP</td><td>6.89</td><td>5</td><td>7.55</td><td>9</td><td>7.11</td><td>7</td></tr>
<tr><td>ResNet-101</td><td>SUP</td><td>6.95</td><td>7</td><td>7.47</td><td>8</td><td>7.21</td><td>8</td></tr>
<tr><td>ResNet-152</td><td>SUP</td><td>7.47</td><td>10</td><td>7.00</td><td>7</td><td>7.47</td><td>9</td></tr>
<tr><td>ViT/L16</td><td>SUP</td><td>7.05</td><td>8</td><td>8.50</td><td>10</td><td>9.11</td><td>10</td></tr>
<tr><td>ViT/S16</td><td>SUP</td><td>10.26</td><td>11</td><td>10.61</td><td>11</td><td>10.37</td><td>11</td></tr>
<tr><td>ViT/S16</td><td>MAE</td><td>11.53</td><td>12</td><td>10.63</td><td>12</td><td>10.42</td><td>12</td></tr>
</table>

Table 1: Comparison of model selection using linear evaluation, combination of evaluation with MLPs and MDL with readout model switching. Average rank (lower is better) for representations over 19 VTAB datasets is reported. Architecture + pre-training objectives are in order of MDL rank.

Table 1a reports the average rank of the pre-training objectives comparing model selection with different evaluation methods. As baselines, we report linear evaluation and a combination of MLPs up to 7-layers. To combine results from MLPs, we take the best performance achieved by MLPs. Given a confidence interval $\gamma = 0.1$, the performance of the representations are significantly different if the difference is above $q_{0.1}\sqrt{r(r+1)/6N} = 1.894$, where $q_{0.1} = 3.12$ is the critical value for the two-tailed Nemenyi test, and $r = 6$ is the number of representations to compare and $N = 19$ is the number of datasets.

Despite fixing the backbone architecture, the choice of readout model has an impact on model selection under accuracy-based evaluations (first two methods in Table 1a). This is consistent with previous observations (Resnick et al., 2019; He et al., 2021) and demonstrates the importance of incorporating a multitude of readout methods into the evaluation framework. Our method solves this issue by combining probing protocols and taking into account their complexity. Furthermore, under our evaluation framework the performance in the small data regime is also an important factor and representations performing worse with little data are penalized in the final rank. We observe that for ResNet50 backbone, BYOL features outperforms both SUPervised and SimCLR features, while SimCLR and SUPervised features have no significant difference. For ViT/B16 backbone, DINO features outperform both supervised and MAE, while supervised and MAE have no significant difference. Therefore, it appears that with the same encoder capacity, distillation-based objectives have an advantage in terms of transfer. DINO seems slightly better than BYOL, but the rank difference is not statistically significant for a clear conclusion.

In Figure 7, we investigate the transfer performance under distribution shift by reporting statistics under each VTAB group. We observe that self-supervised representations have a better performance

than the supervised counterpart on structured datasets, despite the latter perform decently on natural datasets. MAE generalizes well with dataset shift, but the performance on natural datasets suffers compared to other methods.

| Architecture | Obj. | Natural | # | Special | # | Structured | # |
|---|---|---|---|---|---|---|---|
| ViT/B16 | DINO | 1.57 | 1 | 1.00 | 1 | 2.63 | 3 |
| ResNet50 | BYOL | 2.71 | 2 | 3.00 | 2 | 2.13 | 1 |
| ViT/B16 | MAE | 5.71 | 6 | 3.25 | 3 | 2.38 | 2 |
| ViT/B16 | SUP | 2.86 | 3 | 3.25 | 4 | 5.75 | 6 |
| ResNet50 | SimCLR | 4.29 | 5 | 5.75 | 6 | 3.38 | 4 |
| ResNet50 | SUP | 3.86 | 4 | 4.75 | 5 | 4.75 | 5 |

Table 2: Average rank on natrual, special and structured categories of VTAB datasets. The thresholds for the rank difference to be significant are 3.12, 2.92 and 4.13 respectively. Architecture + pre-training objectives are in order of MDL rank on all VTAB tasks.

**Model Scaling.** Another interesting aspect about architecture choice is the model scaling: do larger models result in better performance? We again use ResNet and ViT backbones pre-trained on ImageNet and evaluate on VTAB, but with increasing model sizes. For ResNet, we select ResNet-50, ResNet-101 and ResNet-152, and pre-train with either SUPervised or BYOL objective. For ViT, we use ViT/S16, ViT/B16 and ViT/L16, and pre-train with either SUPervised or MAE objective. We rank the performance of the 12 representations and report the average ranks in Table 1b. With confidence interval of 0.1, we obtain a threshold of 3.65 for the difference in average rank to make a statistical claim.

The scaling experiment more compellingly demonstrates the difference of multiple readout models and the emphasis on performance in the small data regime with our method (Table 1b). Larger models, such as ResNet-152 and ViT/L16, which perform better with MLP head need to demonstrate their performance in the small data regime to justify the additional capacity used for the readout. We can see that scaling up the ResNet architecture, in particular under supervised pre-training, doesn't result in any improvement in the transfer performance. There is a weak positive scaling effect for BYOL when scaling to ResNet-101, but it is not statistically significant. Although ViT/S16 has approximately the same parameter counts as ResNet-50, its predictive performance is much worse, suggesting that ViT's capacity is used less efficiently than ResNet's. Scaling up ViT from S16 to B16 leads to a significant performance improvement. However, further scaling up for supervised loss is not useful. Interestingly, the reconstruction-based MAE objective scales well to larger architectures and has not saturated at ViT/L16.

**Readout Models.** One strength of our evaluation framework is that it can compare readout models of different capacity and dynamically choose the most suitable one based on downstream task and training data size. We compute the posterior over the readout models to inspect which readout model the representation prefers. We observe that model switching is more frequent for out-of-domain datasets than in-domain datasets. In Figure 2, we report the readout model that has most often the highest probability $\arg\max_k \frac{1}{T}\sum_{t=1}^{T} p(\xi_t = k|\mathcal{D}_{<t})$. The result suggests that the most important factor for preferring deeper models is the dataset type - structured datasets need deeper models. The influence of the pre-training objective doesn't seem to be significant.

**Data Efficiency.** Data efficiency is an integral part of MDL. With the prequential perspective on MDL, we can visualize whether an algorithm is more data efficient by inspecting the predictive loss as a function of the data size. As a baseline we train a ResNet-50 model from scratch within our evaluation framework, and record the cumulative next-step losses as a reference

Figure 2: Readout model that most often has the highest probability $\arg\max_k \frac{1}{T}\sum_{t=1}^{T} p(\xi_t = k|\mathcal{D}_{<t})$ on 19 VTAB datasets. 0 is linear readout; 1 to 7 are MLPs with 1 to 7 hidden layers.

| Architecture | Objective | Caltech101 | Cifar100 | DTD | Flowers102 | Pets | SVHN | Sun397 | EuroSAT | PatchCamelyon | Resisc45 | Retinopathy | ClevrCount | ClevrDistance | DMLab | DSpritesLocation | DSpritesOrientation | KittiDistance | SmallNORBAzimuth | SmallNORBElevation |
|---|---|---|---|---|---|---|---|---|---|---|---|---|---|---|---|---|---|---|---|---|
| ResNet50 | SUP | 1 | 1 | 1 | 1 | 1 | 1 | 1 | 1 | 3 | 1 | 3 | 4 | 3 | 3 | 4 | 2 | 4 | 3 | 2 |
| ResNet101 | SUP | 1 | 1 | 3 | 1 | 0 | 1 | 1 | 1 | 2 | 1 | 2 | 3 | 3 | 2 | 2 | 2 | 4 | 3 | 2 |
| ResNet152 | SUP | 1 | 1 | 1 | 2 | 0 | 1 | 1 | 1 | 2 | 1 | 2 | 3 | 4 | 2 | 1 | 2 | 5 | 3 | 7 |
| ResNet50 | SimCLR | 1 | 1 | 1 | 1 | 3 | 2 | 1 | 1 | 4 | 1 | 2 | 7 | 3 | 5 | 3 | 3 | 4 | 7 | 7 |
| ResNet50 | BYOL | 1 | 1 | 1 | 1 | 1 | 2 | 1 | 1 | 6 | 1 | 3 | 4 | 5 | 5 | 4 | 3 | 7 | 6 | 7 |
| ResNet101 | BYOL | 1 | 1 | 1 | 1 | 1 | 2 | 1 | 1 | 3 | 1 | 3 | 4 | 6 | 5 | 3 | 3 | 6 | 6 | 7 |
| ResNet152 | BYOL | 1 | 1 | 1 | 1 | 0 | 2 | 1 | 1 | 3 | 1 | 3 | 5 | 4 | 4 | 2 | 3 | 4 | 6 | 3 |
| ViT/S16 | SUP | 1 | 1 | 1 | 1 | 1 | 1 | 0 | 1 | 4 | 1 | 3 | 3 | 2 | 2 | 4 | 5 | 5 | 7 | 4 |
| ViT/B16 | SUP | 0 | 1 | 2 | 0 | 1 | 1 | 1 | 1 | 3 | 1 | 3 | 7 | 6 | 2 | 3 | 3 | 4 | 3 | 7 |
| ViT/L16 | SUP | 1 | 1 | 2 | 1 | 2 | 0 | 1 | 1 | 3 | 2 | 2 | 7 | 7 | 3 | 1 | 3 | 5 | 4 | 7 |
| ViT/B16 | DINO | 0 | 0 | 1 | 0 | 1 | 1 | 0 | 1 | 3 | 1 | 4 | 5 | 5 | 6 | 3 | 3 | 3 | 3 | 7 |
| ViT/S16 | MAE | 1 | 1 | 1 | 1 | 2 | 2 | 1 | 1 | 3 | 1 | 1 | 2 | 2 | 6 | 7 | 4 | 7 | 6 | 5 |
| ViT/B16 | MAE | 0 | 1 | 1 | 1 | 2 | 1 | 0 | 1 | 2 | 1 | 1 | 3 | 4 | 4 | 4 | 6 | 5 | 7 | 4 |
| ViT/L16 | MAE | 1 | 1 | 1 | 1 | 2 | 1 | 1 | 1 | 2 | 1 | 1 | 4 | 4 | 4 | 4 | 5 | 6 | 5 | 6 |

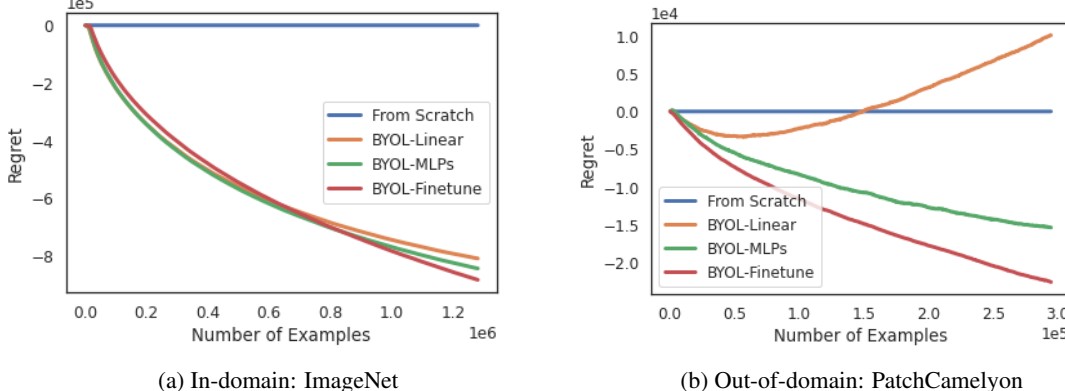

(a) In-domain: ImageNet          (b) Out-of-domain: PatchCamelyon

Figure 3: Data efficiency: We plot the cumulative next-step log-loss for a range of readouts with a ResNet-50 backbone as a function of (downstream) sample size (in nats, lower is better).

baseline (comparator). Then, we evaluate BYOL pretrained ResNet-50 representations with 3 different readout models: 1. linear probing; 2. deep MLP readout models; 3. linear readout but with fine-tuning the whole encoder. Because MDL already takes into account the model complexity, we can directly compare their MDL scores. We plot the "regret" (in this case it is more of an advantage) of different fine-tuning strategies relative to the training-from-scratch model as the baseline. Figure 3 illustrates the "regret" along the data size for ImageNet, an in-domain dataset, and PatchCamelyon, an out-of-domain dataset for the BYOL representations. Learning ImageNet classification with a pre-trained representation is much more efficient than training from scratch. Shallow architectures, which are based on a frozen encoder, are sufficient. Only when the dataset is large enough (800k in case of ImageNet), we can finally see the benefit from fine-tuning the encoder. The picture is different for out-of-domain datasets. In general, pre-training still has benefits if deeper, more flexible readout models are used. The amount of capacity needed depends on the nature of the downstream dataset. In Figure 3, on PatchCamelyon, training from scratch surpasses linear probing after seeing only 150k examples. We show the regret plots for all 19 VTAB tasks in Appendix A.9.

## 5   Conclusions & Future works

We presented readout model switching, an approach derived from the MDL principle, to evaluate representations on given downstream tasks. Our approach supports model switching to choose the most appropriate readout model for each representation, downstream task, and data size. Our metric takes model complexity and data efficiency into account, making it a unified measure for model comparison. We evaluated vision representations and compared our approach to accuracy-based methods. Using our evaluation framework, we were also able to provide insights into model scaling properties and preferred readout models, as well as compare data efficiency.

In theory, different permutations of the data sequence result in different codelengths. Although this does not appear to cause too much trouble in practice (see Appendix A.7), we reckon that this is an area for future work. With the MDL formulation the performance metric under consideration is the log-loss (cross-entropy), which directly corresponds to compression. Extensions to other losses such as 0/1-loss etc. will need careful consideration. We here report a single metric per downstream-task and representation. MDL supports non-stationary sequences and future work could aim for reporting a single metric per sequence-of-tasks by concatenating them. However this poses questions about the order of downstream tasks, and how to deal with multiple output spaces $\mathcal{Y}$, and whether forward transfer between these tasks is permissible.

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

# A APPENDIX

## A.1 PROOF OF REGRET BOUND

Assume that $x \sim P$ and a sequence of examples $x^N = \{x_1, \ldots, x_N\}$ are i.i.d samples from $P$. Let $\hat{\theta}(x^N)$ be the parameters of a readout model $M$ that best fit the sequence $x^N$, i.e. $\hat{\theta}(x^N) = \inf_\theta M(x^N; \theta)$. Since $x^N$ are i.i.d samples from $P$, $\hat{\theta}(x^N)$ is the ML estimator of $x^N$. We assume that all readout models are from exponential family. But we do not assume $P \in M$, i.e. the model can be misspecifed.

Define the expected regret as the regret w.r.t the best sequence of readout head $\xi_{1:N}^*$ and their best fit parameters $\hat{\theta}_{\xi_{1:N}^*}(x^N)$ for $x^N$:

$$
\begin{aligned}
R(n) =& \mathbb{E}_P \left[ \sum_{t=1}^N \log M_{\xi_t^*}(x_t; \hat{\theta}_{\xi_t^*}(x^N)) - \log \sum_{\xi_{1:N}} p(\xi_{1:N}) \prod_{t=1}^N M_{\xi_t}(x_t | \hat{\theta}_{\xi_t}(x^{t-1})) \right] \\
\leq& \mathbb{E}_P \left[ \sum_{t=1}^N \log M_{\xi_t^*}(x_t; \hat{\theta}_{\xi_t^*}(x^N)) - \log p(\xi_{1:N}^*) \prod_{t=1}^N M_{\xi_t^*}(x_t | \hat{\theta}_{\xi_t^*}(x^{t-1})) \right] \\
=& \mathbb{E}_P \left[ \sum_{t=1}^N \left( \log M_{\xi_t^*}(x_t; \hat{\theta}_{\xi_t^*}(x^N)) - \log M_{\xi_t^*}(x_t | \hat{\theta}_{\xi_t^*}(x^{t-1})) \right) - \log p(\xi_{1:N}^*) \right] \\
=& - \mathbb{E}_P \left[ \log p(\xi_{1:N}^*) \right] + \sum_{t=1}^N \mathbb{E}_P \left[ \log M_{\xi_t^*}(x_t; \hat{\theta}_{\xi_t^*}(x^N)) - \log M_{\xi_t^*}(x_t | \hat{\theta}_{\xi_t^*}(x^{t-1})) \right] \quad (4)
\end{aligned}
$$

Since the readout models are from exponential family, they can be parameterized instead by the mean parameters. Recall that the exponential family is the probability distribution that can be written in the form of $P(x; \eta) = h(x) \exp\left(\eta^\top T(x) - A(\eta)\right)$ where $\eta$ is the natural parameters, $T(x)$ is the sufficient statistics, $A(\eta)$ is the cumulant function. The mean parameters are $\mu := \mathbb{E}[T(x)]$ and have a one-to-one mapping to natural parameters $\eta$ (we use $\eta(\mu)$ to denote the mapping). Therefore, the exponential family can also be parameterized by the mean parameters $P(x; \mu) = h(x) \exp\left(\eta(\mu)^\top T(x) - A(\eta(\mu))\right)$. We also know that the sufficient statistics is also an exponential family in the form of $P(T; \mu) = h(T) \exp\left(\eta(\mu)^\top T - A(\eta(\mu))\right)$. Because of the assumption that $M$ is exponential family model, $\mathbb{E}_{x \sim P}[\log M(x; \theta)] = \mathbb{E}_{T \sim P}[\log M(T; \mu)]$. Here, we use the same $P$ as the distribution induced on $T$. The maximum likelihood estimator of the mean parameters is $\hat{\mu}(x^N) = \frac{1}{N} \sum_{t=1}^N T(x_t)$.

Given any estimator $\hat{\mu}$, we can Taylor expand the likelihood at $\mu = \mathbb{E}_P[T(x)]$:

$$
\begin{aligned}
& \log M(x; \hat{\mu}) \\
=& \log M(x; \mu) + (\hat{\mu} - \mu)^\top \nabla_{\hat{\mu}} \log M(x, \hat{\mu})|_\mu + \frac{1}{2}(\hat{\mu} - \mu)^\top \nabla_{\hat{\mu}}^2 \log M(x, \hat{\mu})|_\mu (\hat{\mu} - \mu) + O(1) \\
\implies& \mathbb{E}_P \left[ \log M(x; \hat{\mu}) - \log M(x; \mu) \right] \\
=& \frac{1}{2}(\hat{\mu} - \mu)^\top \mathbb{E}_P \left[ \nabla_{\hat{\mu}} \log M(x, \hat{\mu})|_\mu \right] (\hat{\mu} - \mu) \\
=& \frac{1}{2}(\hat{\mu} - \mu)^\top \mathcal{I}(\mu)(\hat{\mu} - \mu) \\
=& \frac{1}{2}(\hat{\mu} - \mu)^\top \text{Cov}_{M_\mu}^{-1}[T(x)](\hat{\mu} - \mu)
\end{aligned}
$$

Where $\mathcal{I}(\mu)$ is the Fisher information. We first use $\mathbb{E}_P\left[\nabla_{\hat{\mu}} \log M(x, \hat{\mu})\right]|_\mu = \nabla_{\hat{\mu}} \mathbb{E}_P[\log M(x, \hat{\mu})]|_\mu = 0$. Then we use $\mathcal{I}(\mu) = \text{Cov}_{M_\mu}^{-1}[T(x)]$, RHS is the covariance

matrix under the distribution $M_\mu$. For a readout model $M$, after seeing $x^t$ examples, we have

$$\mathbb{E}_{x^{t-1} \sim P} \left[ \log M(x_t; \hat{\mu}(x^{t-1})) - \log M(x_t; \mu) \right]$$

$$= \mathbb{E}_P \left[ \frac{1}{2} (\hat{\mu} - \mu)^\top \text{Cov}_{M_\mu}^{-1} [T(x)] (\hat{\mu} - \mu) \right]$$

$$= \frac{1}{2} \mathbb{E}_P \left[ Tr \left( \text{Cov}_{M_\mu}^{-1} [T(x)] (\hat{\mu} - \mu)(\hat{\mu} - \mu)^\top \right) \right]$$

$$= \frac{1}{2} Tr \left( \text{Cov}_{M_\mu}^{-1} [T(x)] \mathbb{E}_P \left[ (\hat{\mu} - \mu)(\hat{\mu} - \mu)^\top \right] \right)$$

$$= \frac{1}{2} \frac{1}{t-1} Tr \left( \text{Cov}_{M_\mu}^{-1} [T(x)] \text{Cov}_P [T(x)] \right)$$

We use the fact that trace is a linear operator, therefore we could exchange the order with expectation. The last step comes from $\hat{\mu}(x^{t-1}) = \frac{1}{t-1} \sum_i T(x_i)$; denote $T_i = T(x_i)$, we have
$\mathbb{E}_P \left[ (\hat{\mu} - \mu)(\hat{\mu} - \mu)^\top \right] = \mathbb{E}_P \left[ (\frac{1}{t-1} \sum_i T_i - \mu)(\frac{1}{t-1} \sum_i T_i - \mu)^\top \right] = \frac{1}{t-1} \mathbb{E}_P[TT^\top] - \frac{1}{t-1} \mu\mu^\top = \frac{1}{t-1} \text{Cov}_P[T(x)]$. Note here the covariance is under the true data distribution $P$.

For each term inside the sum over timestep $t$ in Equation (4), we have

$$\mathbb{E}_P \left[ \log M_{\xi_t^*}(x_t; \hat{\theta}_{\xi_t^*}(x^N)) - \log M_{\xi_t^*}(x_t | \hat{\theta}_{\xi_t^*}(x^{t-1})) \right]$$

$$= \mathbb{E}_P \left[ \log M_{\xi_t^*}(x_t; \hat{\mu}_{\xi_t^*}(x^N)) - \log M_{\xi_t^*}(x_t | \hat{\mu}_{\xi_t^*}(x^{t-1})) \right]$$

$$= \mathbb{E}_P \left[ \log M_{\xi_t^*}(x_t; \hat{\mu}_{\xi_t^*}(x^N)) - \log M_{\xi_t^*}(x_t; \mu) + \log M_{\xi_t^*}(x_t; \mu) - \log M_{\xi_t^*}(x_t | \hat{\mu}_{\xi_t^*}(x^{t-1})) \right]$$

$$= -\frac{1}{2} (\frac{1}{t-1} - \frac{1}{N}) Tr \left( \text{Cov}_{M_{\mu_{\xi_t^*}}}^{-1} [T_{\xi_t^*}(x)] \text{Cov}_P [T_{\xi_t^*}(x)] \right) + O(1)$$

We first change it from parameterized by natural parameters to mean parameters. Then for each term, we can compare it to $\mu = \mathbb{E}_P[T(x)]$.

The total sum of $t = 1, \ldots, T$ in Equation (4) is

$$\sum_{t=1}^N \mathbb{E}_P \left[ \log M_{\xi_t^*}(x_t; \hat{\theta}_{\xi_t^*}(x^N)) - \log M_{\xi_t^*}(x_t | \hat{\theta}_{\xi_t^*}(x^{t-1})) \right]$$

$$\leq \sum_{t=2}^N \frac{1}{2} (\frac{1}{t-1} - \frac{1}{N}) Tr \left( \text{Cov}_{M_{\mu_{\xi_t^*}}}^{-1} [T_{\xi_t^*}(x)] \text{Cov}_P [T_{\xi_t^*}(x)] \right)$$

$$\leq \frac{1}{2} \sum_{t=2}^N (\frac{1}{t-1} - \frac{1}{N}) \max_k \left\{ Tr \left( \text{Cov}_{M_{\mu_k^*}}^{-1} [T_{\xi_t^*}(x)] \text{Cov}_P [T_{\xi_t^*}(x)] \right) \right\}$$

$$\leq \frac{1}{2} \max_k \left\{ Tr \left( \text{Cov}_{M_{\mu_k^*}}^{-1} [T_k(x)] \text{Cov}_P [T_k(x)] \right) \right\} \left( \sum_{t=2}^N \frac{1}{t-1} \right)$$

$$\leq \frac{1}{2} \max_k \left\{ Tr \left( \text{Cov}_{M_{\mu_k^*}}^{-1} [T_k(x)] \text{Cov}_P [T_k(x)] \right) \right\} \log N$$

Inequality in line 2 uses that When $t = 1$, we use the prior and incur a constant regret. Inequality in line 5 uses harmonic series $\sum_{i=1}^N \frac{1}{i} = \log N + const$. Note that if there is no model misspecification in any of the readout models, we have $M_{\mu_k^*} = P$. In this case, $\text{Cov}_{M_{\mu_k^*}}^{-1} [T_k(x)] \text{Cov}_P [T_k(x)] = I$ and $Tr(I) = k$. We find the classical result for regret of prequential code $\frac{k}{2} \log N$.

Combining with Equation (4), the expected regret is then

$$R(n) \leq -\mathbb{E}_P \left[ \log p(\xi_{1:N}^*) \right] + \frac{1}{2} \max_k \left\{ Tr \left( \text{Cov}_{M_{\mu_k^*}}^{-1} [T_k(x)] \text{Cov}_P [T_k(x)] \right) \right\} \log N \qquad (5)$$

The expected regret for our readout model switching is composed of 2 terms: one is the regret from expert switching algorithm and the other is the regret of the readout models. The growth of second

term is well behaved and bounded by $C \log N$ where $C$ is a constant that relates to the model family chosen. The first term is not necessarily bounded by $\log N$, but relates to the switching strategy chosen. However, for many switching algorithms, such as Bayesian mixture, fixed share with decreasing switching rate and run-length code, $-\mathbb{E}_P\left[\log p(\xi_{1:N}^*)\right]$ is in the order of $\log N$. In particular, for fixed share with decreasing switching rate, we have the result of $-\mathbb{E}_P\left[\log p(\xi_{1:N}^*)\right] \le m \log K + (m-1) \log \frac{N}{m-1}$ where $m-1$ is the number of switch points (i.e. $m$ is the number of blocks in which the same expert is used). See Koolen & de Rooij (2013) for proof of the fixed share and more details of the regret bound of other switching strategy. A similar proof is given by Veness et al. (2012) for the particular case of $m = 2$.

## A.2 DISTRIBUTION SHIFT BETWEEN TRAIN AND TEST IN VTAB

Figure 4 shows the top-1 accuracy on train and test subsets with the original VTAB and reshuffled VTAB datasets. For the same dataset, a difference in test set performance between the original and reshuffled datasets suggests data shift between the two versions. KittiDistance have a significant data shift. DMLab, PatchCamelyon and SmallNorbAzimuth also have distribution shift but to a much less degree compared to KittiDistance. To make sure that the dataset is i.i.d, we combine all its subsets, randomly shuffle then partition into the same number of train/test examples. We only take the training subset for online learning, which results in sequence lengths showed in Table 3. To ensure the distribution shift doesn't affect the conclusion, our experiments in the main paper are run with the reshuffled version of the VTAB dataset. However, we provide additional results on the original VTAB datasets in Appendix A.7.

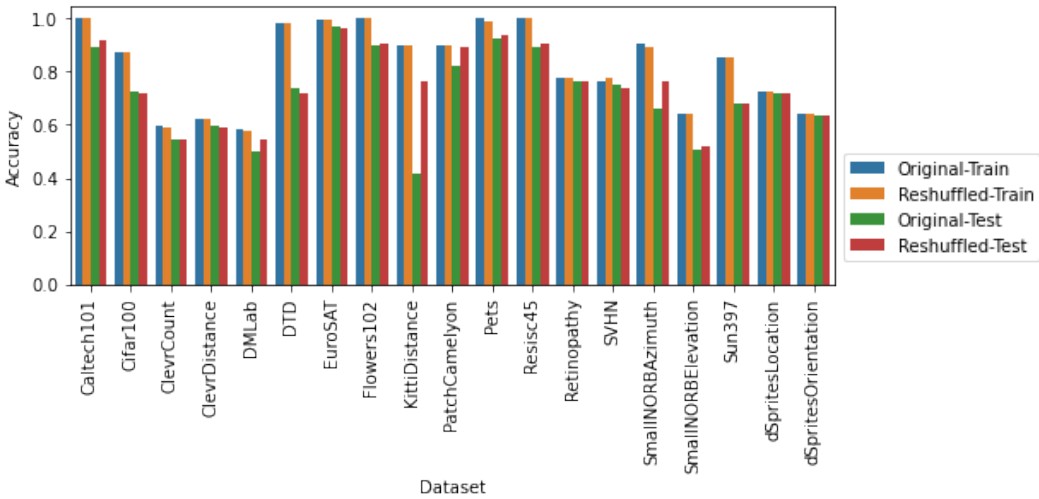

Figure 4: Comparison of train and test accuracies on the original and reshuffled VTAB datasets.

## A.3 2-STAGE ONLINE AND OFFLINE IMPLEMENTATION

The 2-stage implementation is straight forward: first stage, we compute the cross entropy losses for all models at each timestep and store the results; second stage, we use model switching to compute the final codelength (see Algorithm 2).

| Group | Dataset | Number of Examples |
|---|---|---|
| Natural | Caltech101 | 3060 |
| | Cifar100 | 50000 |
| | DTD | 3760 |
| | Flowers102 | 2040 |
| | Pets | 3680 |
| | SVHN | 73257 |
| | Sun397 | 87003 |
| Special | EuroSAT | 21600 |
| | PatchCamelyon | 294912 |
| | Resisc45 | 25200 |
| | Retinopathy | 46032 |
| Structured | ClevrCount | 70000 |
| | ClevrDistance | 70000 |
| | DMLab | 88178 |
| | DSpritesLocation | 663552 |
| | DSpritesOrientation | 663552 |
| | KittiDistance | 6770 |
| | SmallNORBAzimuth | 36450 |
| | SmallNORBElevation | 36450 |

Table 3: Number of examples in the reshuffled VTAB datasets.

---

**Algorithm 2** MDL with Readout Model Switching (2-Stage)

---

Stage 1:

**Require:** data $\mathcal{D} = (x_t, y_t)_{t=1}^T$; $K$ readout models; subroutine that updates the parameters given a dataset *UpdateParameters*

1: Initialize: model parameters $\theta_1, \ldots, \theta_K$; an empty list $L$ to store loss per step and per model
2: **for** $t = 1$ **to** $T$ **do**
3:     Compute next-step loss of $K$ readout models: $\mathcal{L}_t^k := -\log p_k(y_t|x_t) = -\log p(y_t|x_t; \theta_k)$
4:     Store $\mathcal{L}_t^1, \ldots, \mathcal{L}_t^K$ to $L$
5:     **for** $k = 1$ **to** $K$ **do**
6:         Update parameters: $\theta_k \leftarrow UpdateParameters(\theta_k, \mathcal{D}_{\leq t})$
7:     **end for**
8: **end for**
9: **return** $L$

Stage 2:

**Require:** Cross entropy results from stage 1 $L$; initial model probability $p(\xi_1)$ and switching strategy $p(\xi_t|\xi_{t-1})$

1: Initialize: $s = \log p(\xi_1)$; an empty list $Q$ to store posterior $p(\xi_t|\mathcal{D}_{<t})$
2: **for** $t = 1$ **to** $T$ **do**
3:     Compute $\log p(\xi_t, y^{t-1}|x^{t-1})$: $s \leftarrow \log \sum_{\xi_{t-1}} \exp\left(\log p(\xi_t|\xi_{t-1}) + s\right)$
4:     Compute posterior $p(\xi_t|\mathcal{D}_{<t})$ and store in $Q$:
    $\log p(\xi_t|\mathcal{D}_{<t}) = \log p(\xi_t, y^{t-1}|x^{t-1}) - \log \sum_{\xi_t} p(\xi_t, y^{t-1}|x^{t-1})$
5:     Get $\mathcal{L}_t^1, \ldots, \mathcal{L}_t^K$ from $L$ and combine $K$ models to update $\log p(\xi_t, y^t|x^t)$: $s \leftarrow -\mathcal{L}_t^{\xi_t} + s$
6: **end for**
7: Compute total codelength $\mathcal{L}_{switch} \leftarrow -\log \sum_{\xi_T} \exp(s)$
8: **return** $\mathcal{L}_{switch}$ and $Q$

---

### A.4 ACCURACY-BASED EVALUATIONS

For both linear and a combination of MLP evaluations, we perform a hyperparameter search on learning rate, weight decay and EMA with a batch size 1024 (see Table 4 for details). We split the dataset into training and validation where the validation set contains 10% of the data. We choose the best performing hyperparameter on validation set and report top-1 accuracy of the test set in

Table 1a and Table 1b. We treat the readout models type as one of the hyperparameters, choose the one with the lowest loss on validation set and report the test set accuracy.

| Parameter | Values |
|---|---|
| Batch size | 1024 |
| Learning rate | { 1e-4, 3e-4, 1e-3, 3e-3 } |
| Weight decay | { 0.0, 1e-6, 1e-4, 1e-2 } |
| EMA step size (Polyak averaging) | { 1e-4, 1e-2, 1.0 } |

Table 4: Hyperparameters for accuracy-based evaluations on VTAB.

| Architecture | Obj. | Acc (%) | # | Rank | # |
|---|---|---|---|---|---|
| ViT/B16 | DINO | 76.01 | 2 | 2.26 | 1 |
| ResNet-50 | BYOL | 77.93 | 1 | 2.37 | 2 |
| ResNet-50 | SUP | 75.51 | 3 | 3.82 | 3 |
| ViT/B16 | MAE | 74.10 | 5 | 3.84 | 4 |
| ViT/B16 | SUP | 72.46 | 6 | 4.18 | 5 |
| ResNet-50 | SimCLR | 74.35 | 4 | 4.53 | 6 |

Table 5: Comparison of average accuracy v.s. average ranking for summarizing performance on multiple datasets (19 datasets from VTAB) using linear evaluation protocol.

**Average accuracy v.s. Ranking on multiple datasets.** Table 5 shows the difference on model selection when using average accuracy and average ranking for aggregating performances on multiple datasets. We see that the two summarization methods don't have the same results for model selection. The reason why it's problematic to average accuracy over multiple datasets is discussed in Demšar (2006). In brief, averaging over accuracy is only meaningful when the results on different datasets are comparable and averages are susceptible to outliers. Since the VTAB benchmark consists of 19 datasets for classification tasks of very different nature, averaging over the accuracy is an ill-conceived practice and we choose to use average ranking for presenting our results.

## A.5 Implementation Details

### A.5.1 VTAB

**Frozen Encoders** For frozen encoders, we use the readout models: linear layer and MLP of 1-7 hidden layers. We use AdamW as the optimizer and hyperparameters listed in Table 6. The switching probability for fix share if $\alpha_t = 10/t$. Images are center cropped and resized to $224 \times 224$.

| Parameter | Values |
|---|---|
| Batch size | 32 |
| Number of replay-streams | { 3, 10, 30, 100 } |
| Learning rate | { 1e-4, 3e-4, 1e-3, 3e-3 } |
| AdamW $\beta_1$ | { 0.5, 0.7, 0.9 } |
| Weight decay | { 0.0, 1e-6, 1e-4, 1e-2 } |
| EMA step size (Polyak averaging) | { 1e-4, 1e-2, 1.0 } |

Table 6: Hyperparameters for computing MDL of frozen encoders on VTAB.

**Train from Scratch** This section correspond to the training from scratch experiment on VTAB presented in Section 4. We use SGD with nesterov momentum for training from scratch. We take the center crop and resize the image to $224 \times 224$ for preprocessing. No readout models is needed

as the encoder is trained on the relevant task. For each VTAB task, we uniformly randomly select 300 configurations from the hyperparameters listed in Table 7.

| Parameter | Values |
|---|---|
| Batch size | 32 |
| Number of replay-streams | { 50, 100, 150, 200 } |
| Learning rate | { 0.001, 0.003, 0.01, 0.03, 0.1, 0.3, 1.0 } |
| momentum | 0.9 |
| Weight decay | { 1.e-5, 3e-5, 1e-4, 3e-4, 1e-3, 3e-3, 1e-2, 3e-2 } |
| EMA step size (Polyak averaging) | { 1e-3, 1e-2, 1e-1, 1.0 } |

Table 7: Hyperparameters for computing MDL with training ResNet-50 from scratch on VTAB.

**Fine-tuning** This section correspond to the fine-tuning experiment on VTAB presented in Section 4. We use ResNet-50 encoder for the fine-tuning experiment. Readout model is a linear layer. The encoder parameters are initialized with the pre-trained parameters of BYOL, but updated on the downstream task. We use SGD with nesterov momentum for fine-tuning pre-trained encoders. Images are center cropped and resized to $224 \times 224$ for preprocessing. For each VTAB task, we uniform randomly select 300 configurations from the hyperparameters listed in Table 8.

| Parameter | Values |
|---|---|
| Batch size | 32 |
| Number of replay-streams | { 10, 30, 50 } |
| Learning rate | { 1e-4, 3e-4, 1e-3, 3e-3, 0.01, 0.03, 0.1, 0.3, 1. } |
| momentum | 0.9 |
| Weight decay | { 0.0, 1.e-5, 3e-5, 1e-4, 3e-4, 1e-3, 3e-3 } |
| EMA step size (Polyak averaging) | { 1e-3, 1e-2, 1e-1, 1.0 } |

Table 8: Hyperparameters for computing MDL with fine-tuning pre-trained encoders on VTAB.

### A.5.2 IMAGENET

**Frozen Encoders** Images are preprocessed with spatial augmentation during training: randomly cropped, randomly flipped and resized to $224 \times 224$. We use the readout models: linear layer and MLP of 1-3 hidden layers. We use AdamW as the optimizer and hyperparameters listed in Table 9. The switching probability for fix share if $\alpha_t = 1/t$.

| Parameter | Values |
|---|---|
| Batch size | 512 |
| Number of replay-streams | { 10, 30, 50 } |
| Learning rate | { 3e-5, 1e-4, 3e-4, 1e-3 } |
| AdamW $\beta_1$ | { 0.5, 0.7 } |
| Weight decay | { 1e-2, 1. } |
| EMA step size (Polyak averaging) | { 0.01, 1. } |

Table 9: Hyperparameters for computing MDL of frozen encoders on ImageNet.

**Fine-tuning** This section correspond to the fine-tuning experiment on ImageNet presented in Section 4. We use a ResNet-50 encoder for the fine-tuning experiment. Readout model is a linear layer. The encoder parameters are initialized with the pre-trained parameters from BYOL, but updated on the downstream task. We use AdamW and spatial augmentation for preprocessing. Images are randomly cropped and resized to $224 \times 224$. We use the same set of hyperparameters as the frozen encoder experiments listed in Table 9.

**Train from Scratch**  We train a standard ResNet-50 architecture from scratch on ImageNet and follow the experimental setting described in (Bornschein et al., 2022). In particular we use replay-streams training with uniform replay, AdamW, use randaugment for data augmentation, and forward calibration. We run a random hyperparameter search by sampling 50 hyperparameter configurations from the intervals listed in appendix A.5.2.

| Parameter | Distribution | Values / Interval |
|---|:---:|---|
| Batch size | fixed | 512 |
| Number of replay-streams | log-uniform | $25 \ldots 100$ |
| Learning rate | log-uniform | 1e-4 $\ldots$ 3e-3 |
| AdamW $\epsilon$ | log-uniform | 1e-4 $\ldots$ 1 |
| EMA step size (Polyak averaging) | log-uniform | 1e-3 $\ldots$ 1e-2 |
| Weight decay | log-uniform | 1e-4 $\ldots$ 1. |
| Label smoothing | fixed | 0.01 |

### A.6 COMPARISON OF SWITCHING STRATEGIES

We describe briefly other switching strategy we experimented with our evaluation framework:

- Bayesian mixture: switching probability is 1 after the first step.

$$L_{BM}(y_{1:T}|\phi) = -\log \sum_k p(k) \prod_{t=1}^{T} p(y_t|\phi_t, \hat{\theta}_k(\phi_{<t}, y_{<t}))$$

- Elementwise mixture: at each step, it is allowed to switch to another model with a fixed probability which doesn't depend on the current model used.

$$L_{EW}(y_{1:T}|\phi) = -\log \sum_{\xi_{1:T}} \prod_{t=1}^{T} p(y_t|\phi_t, \hat{\theta}_{\xi_t}(\phi_{<t}, y_{<t})) w(\xi_t)$$

  Where $w(\xi_t)$ is a fixed mixture weight which doesn't depend on $\xi_{t-1}$.

- Switch distribution: A more sophisticated switching strategy which augments the transition with two latent variables $u$ and $s$ for unstable and stable chain respectively. Once switching to the stable chain, it stay with the same model for the rest of the sequence. For more details, see Koolen & De Rooij (2008).

Figure 5 compares the above switching strategies with the fixed share with decreasing switching rate on ImageNet using a ResNet50 pre-trained with BYOL. Bayesian mixture and elementwise mixture perform worse than fixed share by a large margin. Switch distribution is very close to fixed share performance.

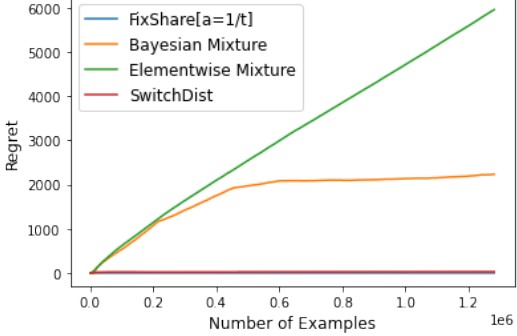

Figure 5: Regret of Bayesian mixture, elementwise mixture and switching distribution compared to the fixed share baseline on ImageNet. Pretrained model is BYOL with ResNet50 backbone.

### A.7  MDL ON VTAB DATASETS

We report the final MDL score using fixed share switching strategy on all the VTAB datasets in Table 10. To better compare the numbers, we average the MDL score by the number of examples in the datasets. For cumulative MDL score, we should multiple by the number of examples which is reported in Table 3.

| Architecture | Objective | Natural | | | | | | | Special | | | | Structured | | | | | | | |
|---|---|---|---|---|---|---|---|---|---|---|---|---|---|---|---|---|---|---|---|---|
| | | Caltech101 | Cifar100 | DTD | Flowers102 | Pets | SVHN | Sun397 | EuroSAT | PatchCamelyon | Resisc45 | Retinopathy | ClevrCount | ClevrDistance | DMLab | DSpritesLocation | DSpritesOrientation | KittiDistance | SmallNORBAzimuth | SmallNORBElevation |
| ResNet-50 | SUP | 0.81 | 1.12 | 1.35 | 1.23 | 0.35 | 0.82 | 1.42 | 0.14 | 0.17 | 0.45 | 0.72 | 0.91 | 0.87 | 0.90 | 0.14 | 0.26 | 0.58 | 0.51 | 0.88 |
| ResNet-50 | SimCLR | 0.98 | 1.33 | 1.33 | 1.30 | 0.74 | 0.66 | 1.54 | 0.18 | 0.20 | 0.46 | 0.73 | 0.90 | 0.79 | 0.92 | 0.08 | 0.25 | 0.58 | 0.29 | 0.77 |
| ResNet-50 | BYOL | 0.78 | 0.97 | 1.21 | 1.06 | 0.48 | 0.69 | 1.32 | 0.13 | 0.17 | 0.39 | 0.70 | 0.78 | 0.77 | 0.77 | 0.13 | 0.16 | 0.55 | 0.24 | 0.65 |
| ResNet-101 | SUP | 0.76 | 1.02 | 1.31 | 1.27 | 0.34 | 0.95 | 1.38 | 0.14 | 0.18 | 0.48 | 0.73 | 0.89 | 0.91 | 0.91 | 0.13 | 0.23 | 0.59 | 0.52 | 0.92 |
| ResNet-101 | BYOL | 0.72 | 0.86 | 1.17 | 1.04 | 0.43 | 0.78 | 1.25 | 0.13 | 0.17 | 0.38 | 0.70 | 0.77 | 0.79 | 0.77 | 0.11 | 0.16 | 0.55 | 0.24 | 0.63 |
| ResNet-152 | SUP | 0.74 | 0.99 | 1.32 | 1.28 | 0.34 | 0.99 | 1.37 | 0.15 | 0.18 | 0.50 | 0.72 | 0.90 | 0.94 | 0.91 | 0.18 | 0.23 | 0.60 | 0.50 | 0.92 |
| ResNet-152 | BYOL | 0.71 | 0.84 | 1.15 | 1.07 | 0.42 | 0.79 | 1.22 | 0.13 | 0.17 | 0.40 | 0.70 | 0.77 | 0.77 | 0.78 | 0.10 | 0.15 | 0.55 | 0.25 | 0.66 |
| ViT/S16 | SUP | 0.83 | 0.95 | 1.45 | 1.45 | 0.38 | 1.23 | 1.38 | 0.19 | 0.20 | 0.56 | 0.72 | 1.06 | 1.17 | 1.14 | 0.63 | 0.38 | 0.64 | 0.61 | 1.00 |
| ViT/S16 | MAE | 1.83 | 2.01 | 1.76 | 2.10 | 1.54 | 1.64 | 2.27 | 0.17 | 0.15 | 0.64 | 0.71 | 0.97 | 1.16 | 1.09 | 1.75 | 0.27 | 0.64 | 0.59 | 0.69 |
| ViT/B16 | SUP | 0.71 | 0.82 | 1.35 | 1.17 | 0.32 | 0.99 | 1.28 | 0.14 | 0.17 | 0.43 | 0.70 | 0.94 | 1.09 | 1.01 | 0.35 | 0.25 | 0.60 | 0.47 | 0.88 |
| ViT/B16 | DINO | 0.68 | 0.68 | 1.14 | 0.94 | 0.37 | 0.76 | 1.15 | 0.11 | 0.13 | 0.34 | 0.68 | 0.78 | 0.94 | 0.72 | 0.17 | 0.19 | 0.55 | 0.28 | 0.57 |
| ViT/B16 | MAE | 1.12 | 1.21 | 1.49 | 1.64 | 0.93 | 0.93 | 1.60 | 0.13 | 0.14 | 0.47 | 0.70 | 0.70 | 0.80 | 0.92 | 0.03 | 0.14 | 0.58 | 0.29 | 0.48 |
| ViT/L16 | SUP | 0.81 | 0.88 | 1.52 | 1.41 | 0.37 | 1.06 | 1.40 | 0.18 | 0.18 | 0.52 | 0.72 | 0.96 | 1.05 | 1.04 | 0.32 | 0.27 | 0.61 | 0.59 | 0.91 |
| ViT/L16 | MAE | 0.92 | 0.86 | 1.39 | 1.50 | 0.53 | 0.82 | 1.36 | 0.12 | 0.13 | 0.43 | 0.69 | 0.60 | 0.76 | 0.85 | 0.02 | 0.11 | 0.56 | 0.21 | 0.42 |

Table 10: Fixed share codelength per example reported for each VTAB dataset.

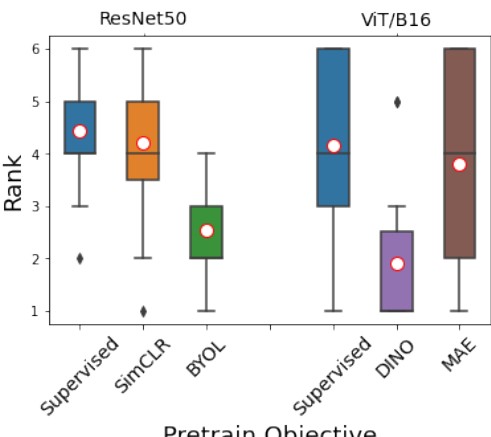

Figure 6: Ranking for representations pre-trained with different objectives (lower the better). The box plot is drawn from 25% to 75% quantile with a horizontal line to denote the median and red dot to denote the mean. The whisker denotes 1.5 IQR values. The threshold of the ranking difference to be significant for hypothesis testing is $1.894$.

In VTAB, datasets in the same group are similar to each other, but there is a significant distribution shift between groups. And natrual datasets are closer to the ImageNet which is used during pre-training. To investigate if the transfer performance differs with input distribution shift, we report the average rankings by dataset group in Figure 7. Although the number of datasets in each group is limited and, therefore, the result from hypothesis test is not conclusive, we can still get some idea of the transfer performance from the ranking difference. We can see that self-supervised representations have a better performance than the supervised counterpart on structured datasets, despite the latter perform decently on natural datasets. MAE generalizes well with dataset shift, but the performance on natural datasets suffers compared to other methods.

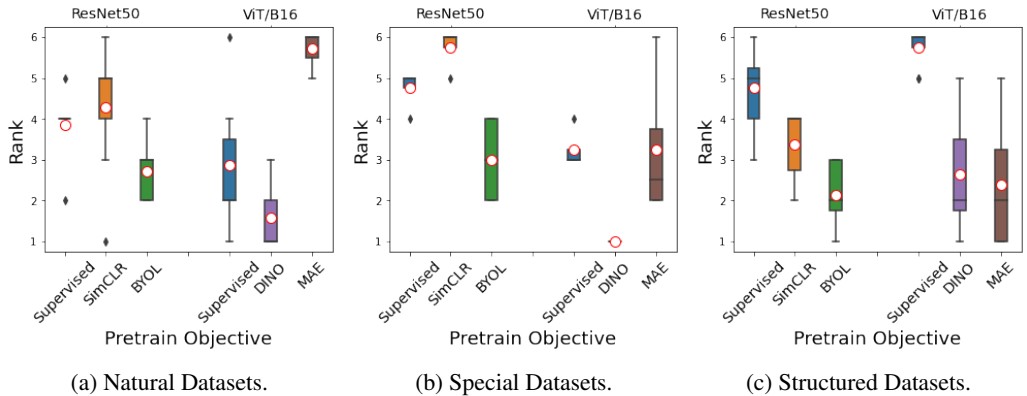

(a) Natural Datasets.  (b) Special Datasets.  (c) Structured Datasets.

Figure 7: Ranking on natrual, special and structured VTAB datasets. The thresholds for ranking difference to be significant are $3.12$, $2.92$ and $4.13$ respectively.

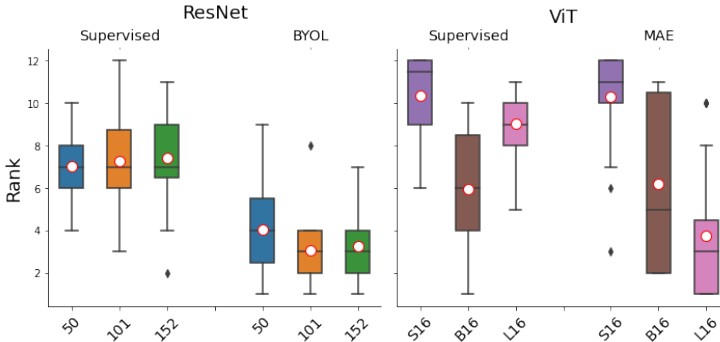

Figure 8: Rankings of ResNet and ViT models of different sizes (lower the better). The thresholds for ranking difference to be significant is $3.65$.

**Data ordering and Variance of MDL score.** Theoretically, different permutations of the data sequence results in different codelengths. We reckon that this is an area for future work. Rissanen (1986) suggests taking the permutation with the shortest codelength, but it is computationally too expensive. Hansen & Yu (2001) notes that the codelength is invariant over orderings when a Bayesian predictor is used in prequential MDL. De Rooij & Grünwald (2011) points out that while the ordering of data affects the codelength, it does not appear to cause too much trouble in practice. Here, we provide an empirical evaluation of the variance of our metric. We evaluate ResNet50-SUP representation on Cifar100 with 5 permutations and obtain the MDL score of 55906 with a standard deviation of 134. To give an idea, the representations that are ranked immediately above and below achieve the MDL score of 50969 and 60549 respectively. Therefore, we believe our evaluation method is stable and reliable. We would also like to point out that the current practice of comparing the test accuracy is not immune to randomness, e.g. initialization, dataset split and randomization during training.

**MDL Results on original VTAB Datasets** Table 11 reports the MDL score on the original VTAB datasets, as well as the average rank. The distribution shift results in changes of the MDL score (colored in blue and red in Table 11). The datasets where the distribution shift occur are consistent with our previous assessment in Appendix A.2. However, the overall rank of the pre-training method doesn't seem to be affected. This is because the number of datasets affected by the shift is relatively small and the effect of the shift is similar to all pre-training methods.

| Architecture | Objective | # | Avg Rank | Natural | | | | | | | Special | | | | Structured | | | | | | | |
|---|---|---|---|---|---|---|---|---|---|---|---|---|---|---|---|---|---|---|---|---|---|---|
| | | | | Caltech101 | Cifar100 | DTD | Flowers102 | Pets | SVHN | Sun397 | EuroSAT | PatchCamelyon | Resisc45 | Retinopathy | ClevrCount | ClevrDistance | DMLab | DSpritesLocation | DSpritesOrientation | KittiDistance | SmallNORBAzimuth | SmallNORBElevation |
| ResNet-50 | SUP | 4 | 4.26 | 0.85 | 1.12 | 1.35 | 1.24 | 0.33 | 0.84 | 1.42 | 0.14 | 0.15 | 0.45 | 0.71 | 0.91 | 0.87 | 0.82 | 0.14 | 0.26 | 0.56 | 0.49 | 0.85 |
| ResNet-50 | SimCLR | 6 | 4.31 | 1.08 | 1.34 | 1.35 | 1.29 | 0.68 | 0.67 | 1.55 | 0.18 | 0.18 | 0.45 | 0.72 | 0.90 | 0.81 | 0.83 | 0.08 | 0.25 | 0.56 | 0.30 | 0.74 |
| ResNet-50 | BYOL | 2 | 2.42 | 0.82 | 0.98 | 1.23 | 1.07 | 0.46 | 0.70 | 1.33 | 0.13 | 0.15 | 0.38 | 0.69 | 0.78 | 0.77 | 0.69 | 0.13 | 0.16 | 0.54 | 0.26 | 0.62 |
| ViT/B16 | SUP | 4 | 4.26 | 0.71 | 0.83 | 1.36 | 1.19 | 0.30 | 1.02 | 1.29 | 0.14 | 0.15 | 0.42 | 0.69 | 0.94 | 1.09 | 0.93 | 0.35 | 0.25 | 0.59 | 0.47 | 0.84 |
| ViT/B16 | DINO | 1 | 1.95 | 0.69 | 0.69 | 1.18 | 0.95 | 0.34 | 0.78 | 1.15 | 0.11 | 0.11 | 0.33 | 0.67 | 0.78 | 0.94 | 0.65 | 0.17 | 0.19 | 0.54 | 0.29 | 0.54 |
| ViT/B16 | MAE | 3 | 3.78 | 1.23 | 1.22 | 1.52 | 1.64 | 0.88 | 0.93 | 1.60 | 0.13 | 0.12 | 0.46 | 0.69 | 0.70 | 0.80 | 0.83 | 0.03 | 0.14 | 0.57 | 0.30 | 0.47 |

Table 11: Fixed share codelength per example reported for original VTAB benchmark where some of the datasets have a distribution shift. The number is color coded if the codelength changes more than $1e^{-2}$ compared to the reshuffled version of the dataset. Blue: codelength is longer than reshuffled dataset; red: codelength is shorter.

### A.8 POSTERIOR OF THE READOUT MODELS

Figure 9 shows the posterior of readout models $p(\xi_t = k|\mathcal{D}_{<t})$ where $k \in [1, \ldots, K]$ on ImageNet for 4 different architecture and pre-training methods: ResNet50 + SimCLR, ResNet50 + BYOL, ViT/B16 + DINO and ViT/B16 + MAE. We use 4 readout models: linear layer and MLP with 1-3 hidden layers.

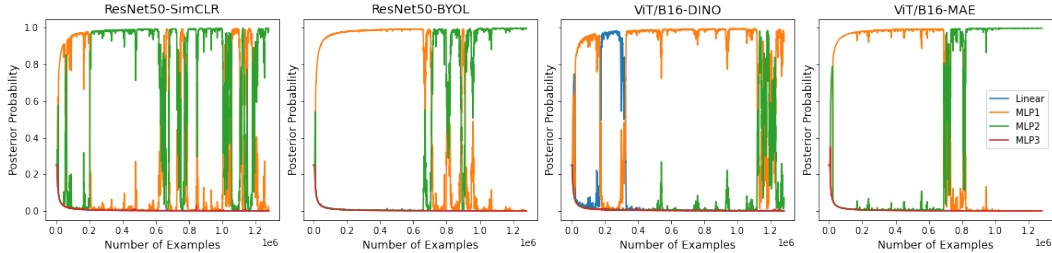

Figure 9: Posterior probability of the readout models at each timestep on ImageNet. Pre-trained methods are ResNet50 + SimCLR, ResNet50 + BYOL, ViT/B16 + DINO and ViT/B16 + MAE

### A.9 DATA EFFICIENCY AND FINE-TUNING

Figure 10 shows the regret plots, relative to training from scratch, for training a linear layer on top of the pre-trained and frozen BYOL backbone, training MLPs (1-7 hidden layers) on top of the frozen BYOL backbone and for a linear readout but while fine-tuning all the encoder parameters. The network architecture is a ResNet-50 in all cases.

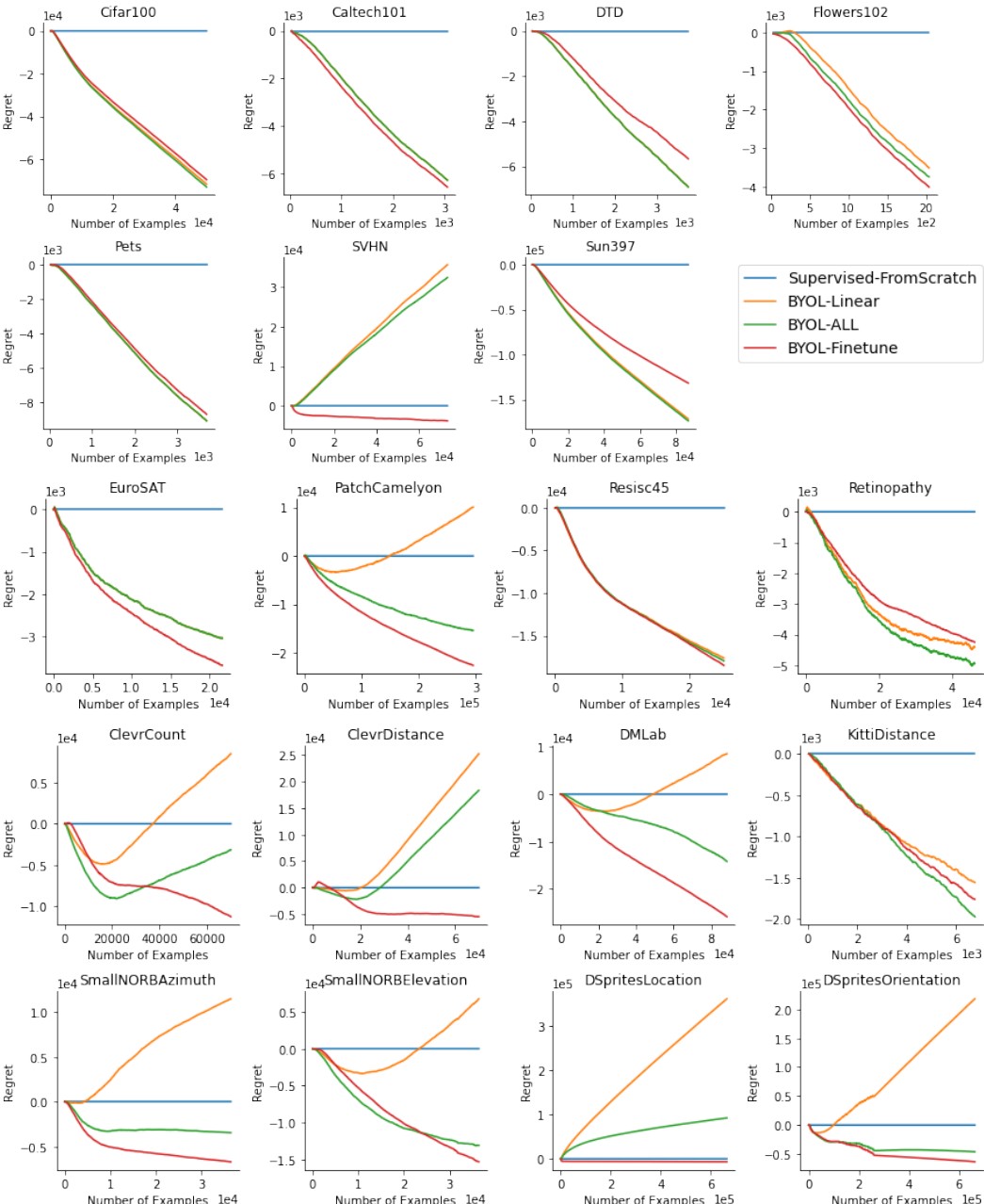

Figure 10: Data efficiency on 19 VTAB datasets. Four methods are compared: training from scratch, training a linear layer with frozen encoder initialized with BYOL, training MLPs with frozen encoder initialized with BYOL and fine-tuning the encoder initialized with BYOL. We plot the cumulative next-step log loss relative to training from scratch as the baseline (in nats; lower is better).

