# OpenReview forum: "Evaluating Representations with Readout Model Switching"
_ICLR.cc/2023/Conference — ICLR 2023 poster_

### Official Review · Reviewer_4XSa · 2022-10-19

**Confidence:** 1
**Correctness:** 3
**Technical Novelty And Significance:** 3
**Empirical Novelty And Significance:** 3
**Recommendation:** 8

**Clarity, Quality, Novelty And Reproducibility:**

-

**Strength And Weaknesses:**

-

**Summary Of The Paper:**

This paper proposes a metric for comparing learned representations, and furthermore doing it so that models can be switched based on data. That sounds useful for a variety of purposes. That said:

This paper was assigned to a completely wrong reviewer; an indication of some sort of a failure in the ICLR process. I don't have the necessary background to appreciate much of this, and reading the paper very carefully would have taken too long -- and probably wouldn't have been too useful for the review process anyway. I have informed the AC that I'll essentially skip the review by putting in a minimum confidence opinion that should not count towards the outcome.

**Summary Of The Review:**

See above.

---

### Official Review · Reviewer_pMpS · 2022-10-23

**Confidence:** 3
**Correctness:** 3
**Technical Novelty And Significance:** 3
**Empirical Novelty And Significance:** 2
**Recommendation:** 6

**Clarity, Quality, Novelty And Reproducibility:**

In this paper, the author proposes a switching policy to select probing models for representation evaluation. It is quite clear for the idea of the paper. I think the major novelty is the authors introducing the switching scheme during the evaluation, not simply applying the MDL. Since the proposed method is time-dependent and the author has re-split the dataset, it would be easier for other researchers to reproduce the results if such details are revealed.

**Strength And Weaknesses:**

Strength:
The author proposes a new method to evaluate the representations of deep learning models based on a selection of different readout (or expert) models according to the MDL. If the author could adequately approve the advantage of the proposed metric, this could bring a new angle for model evaluation.

Weakness:
1. Although the authors have claimed that a rigorous method for representation evaluation is lacking for deep learning models, they didn’t show the advantage of their evaluation methods over others, especially in the experiments. Furthermore, for most unsupervised models, the representation quality is evaluated indirectly by multiple downstream tasks, where the model is required to show consistent improvements on multiple tasks to show its advantage. If the author claims that their method could be more accurate in evaluation, they should at least show some empirical comparisons, where the multi-task evaluation metrics fail but the readout switching method succeeds.

2. The proposed switching method is based on Minimum Description Length, which is used as an alternative method for language model evaluation where the probing tasks fail to reflect differences in representations [1]. To adapt this in the vision task (as the models used in the experiment are vision models), the author should show that the vision model suffers from similar problems, or the MDL could provide a better evaluation solution than current solutions.

3. Even if the author's claim is true, the representation could still be compared by a combination of the performance on several probing models without switching. If the author wants to show the advantage of the proposed switching method, there should be an experiment that compares the evaluation difference of using a) the proposed switching method and b) a combination of probing models without switching.

4. The author claims that different probing models could bring differences during the evaluation of the representation of deep learning models. However, the selection of switching policy may also affect the evaluation metric. If so, what is the principle to find the best selection strategy?

5. According to Algorithm 1, the switching happens at each time step when taking in a new pair of data (x_t, y_t). So what will happen to the proposed metric if the order of evaluation data changes?

6. For the experiment in 4.1, the authors apply different supervision methods on the two models (looks like the “Supervised” is the same for both?). Why not just apply the proposed five supervision methods to both models for comparison?

7. Although the author has listed the evaluation scores of different models and shown the rankings, there remains a question of whether the models behave consistently with this conclusion. For example, in Figure 4, the author shows that a deeper network could be worse than a shallow one (like ResNet, and ViT-B16 vs ViT-L16) in “Supervised”. This conclusion is quite questionable as shown in [2][3]

Writing:
1. For formulas, especially in section 2, it would be easier for the reader to review if the formula could be put in a separate row.
2. In the last paragraph of Regret Bounds, DNN has been defined at the beginning.
3. In 4.1, what does it mean by “pre-train on supervised losses”? Is the supervision method the same for ResNet and ViT?

Reference:

[1] Voita, Elena, and Ivan Titov. "Information-theoretic probing with minimum description length." arXiv preprint arXiv:2003.12298 (2020).

[2] Gao, Shang-Hua, et al. "Res2net: A new multi-scale backbone architecture." IEEE transactions on pattern analysis and machine intelligence 43.2 (2019): 652-662.

[3] Liu, Ze, et al. "Swin transformer: Hierarchical vision transformer using shifted windows." Proceedings of the IEEE/CVF International Conference on Computer Vision. 2021.

[4] Khan, Salman, et al. "Transformers in vision: A survey." ACM computing surveys (CSUR) 54.10s (2022): 1-41.

**Summary Of The Paper:**

This paper provides a method to evaluate the representations of deep learning models by combining predictions of readout models. The readout models are selected based on a switching policy according to the Minimum Description Length, which is an evaluation metric for probing models that consider both model accuracy and complexity. The switching policy is determined by a stochastic matrix that defines the probability of replacing the current expert model. With the proposed method, the author evaluates the representation efficiency of popular deep learning with different scales on a set of downstream tasks and investigates the data efficiency.

**Summary Of The Review:**

This paper proposes an evaluation metric for representations of deep learning models that rely on switching within multiple read-out models based on MDL. The author has compared different models, i.e., ResNet and ViT, based on the proposed metric. However, a major concern is that: Is the proposed method could evaluate representations effectively, or at least better than the current methods? If the authors plan to show the stability and reliability of the proposed methods, there should at least be some experiments that show the failure of compared evaluation metrics, such as by a weighted average over the performance of some downstream tasks. Furthermore, if the authors want to propose an effective evaluation metric, it is necessary to show consistency when selecting different switching strategies.

----Post Rebuttal----

The author has answered most of my questions and the revised version has motivated me to move my rating from 5 -> 6. For me, the author has proposed an effective (or stable) method to evaluate the representation by combining evaluation over different readout models. However, I still have some concerns about the efficiency of the switching policy, which may be left in future works.

---

> ### Author Response · Authors · 2022-11-14
> **Author's response (1/2)**
>
> We thank the reviewer for the thoughtful review and careful consideration of our paper.
>
> > they didn’t show the advantage of their evaluation methods over others, especially in the experiments.
>
> Please see the comparison section in the common response.
>
> > Furthermore, for most unsupervised models, the representation quality is evaluated indirectly by multiple downstream tasks, where the model is required to show consistent improvements on multiple tasks to show its advantage.
>
> We don't see how multiple downstream tasks could help alleviate the issue of different probes being used. We will expand more on the core issue our method help to tackle in the response after the next one.
>
> > If the author claims that their method could be more accurate in evaluation, they should at least show some empirical comparisons, where the multi-task evaluation metrics fail but the readout switching method succeeds.
>
> Please see our common response for the comparisons.
>
> > To adapt this in the vision task (as the models used in the experiment are vision models), the author should show that the vision model suffers from similar problems, or the MDL could provide a better evaluation solution than current solutions.
>
> The core issue of the current evaluation method is that we cannot differentiate the quality of the representation from what's learned by the probes afterward during downstream training. This problem doesn't go away just because we are dealing with vision tasks. In fact, the same problem manifests in vision already: multiple evaluation protocols are used, even new protocols are proposed, and conflicting model selection choices are concluded. We demonstrate this issue in the common response with conflicting model rankings under linear and a combination of MLP evaluations.
>
> > If the author wants to show the advantage of the proposed switching method, there should be an experiment that compares the evaluation difference of using a) the proposed switching method and b) a combination of probing models without switching.
>
> Both points have been addressed in comparisons in the common response. We would like to emphasize that there is no established method to combine probing models.
>
> > However, the selection of switching policy may also affect the evaluation metric. If so, what is the principle to find the best selection strategy?
>
> Like any (prior) choice for modelling, the switching strategy has an effect on the codelength. Two things need to be considered when choosing a strategy: 1) the prior is flexible enough; 2) the regret bound is sublinear. Switching strategies like fixed share and switch-distribution make similar assumptions and empirically we observe that they have very similar results (see Appendix A.5) . Switching strategies with very different assumptions like the Bayesian mixture (assuming a single readout-model is the best throughout the dataset) or element-wise mixture (the best readout-model for example t is unrelated to the model for t+1) lead to significantly worse description lengths. The prior of the Bayesian mixture is likely not flexible enough for the task at hand; element-wise mixture has no guarantee of a sublinear regret bound. In the end, we chose the fixed-share strategy with decreasing switch probability because it matches our prior assumptions (a particular readout model is responsible for a range of examples in a row), has a sublinear regret and because it showed best empirical results in terms of compression.
>
> > So what will happen to the proposed metric if the order of evaluation data changes?
>
> Please see the common response for the order of data.
>
> > For the experiment in 4.1, the authors apply different supervision methods on the two models (looks like the “Supervised” is the same for both?). Why not just apply the proposed five supervision methods to both models for comparison?
>
> We use established supervision methods with their proposed backbone and pre-training scheme. We didn't try combining architectures with arbitrary pre-training losses because we know that the training setup (optimization, batching, data augmentation, etc) is important for the performance of these models.

---

> > ### Author Response · Authors · 2022-11-14
> > **Author's response (2/2)**
> >
> > > Although the author has listed the evaluation scores of different models and shown the rankings, there remains a question of whether the models behave consistently with this conclusion. For example, in Figure 4, the author shows that a deeper network could be worse than a shallow one (like ResNet, and ViT-B16 vs ViT-L16) in “Supervised”. This conclusion is quite questionable as shown in [2][3].
> >
> > We provide ImageNet test set performance of the pre-trained models used in our paper:
> >
> > |Model | Top1 Accuracy(%)|
> > | :- | :------: |
> > |ResNet50-SUP | 76.85|
> > |ResNet101-SUP | 78.62|
> > |ResNet152-SUP | 79.06|
> > |ViT/S16-SUP		|79.90|
> > |ViT/B16-SUP 		|82.20|
> > |ViT/L16-SUP		|81.58|
> > |ResNet50-SimCLR	|70.46|
> > |ResNet50-BYOL	|74.47|
> > |ViT/B16-DINO	|	78.00|
> > |ViT/B16-MAE	|	67.03|
> >
> > Note that, for the ImageNet test set, our ResNet50 and ResNet101 baselines outperform the same baselines provided in [2] (76.15% and 77.37% respectively). We also have a better baseline for ViT models compared to [3] (79.8% and 81.8% for S and B). We cannot find relevant and comparable results in [2] and [3] to support the reviewer’s claim that deeper networks should perform better than the shallow one under the same transfer datasets that we investigate. On the contrary, [1] has already shown, for the same VTAB benchmark, that the performance of ResNet152 is worse than ResNet101.
> >
> > > For formulas, especially in section 2, it would be easier for the reader to review if the formula could be put in a separate row.
> >
> > We are going to improve the presentation of the formula in the paper.
> >
> > > In the last paragraph of Regret Bounds, DNN has been defined at the beginning.
> >
> > We will remove the redundant definition and use the abbreviation.
> >
> > > In 4.1, what does it mean by “pre-train on supervised losses”? Is the supervision method the same for ResNet and ViT?
> >
> > For transfer, we have two training phases: pre-training on ImageNet and downstream training on the transfer dataset. By “pre-train on supervised losses”, we meant that the pre-training phase uses label information and cross entropy loss. For the self-supervision, we only use established self-supervised methods with their choice of the backbone in the original paper, i.e. SimCLR and BYOL with ResNet; DINO and MAE with ViT.
> >
> > > It would be easier for other researchers to reproduce the results if such details are revealed
> >
> > We are going to make the code and datasets available.
> >
> >
> > [1] Zhai, Xiaohua, et al. "A large-scale study of representation learning with the visual task adaptation benchmark." arXiv preprint arXiv:1910.04867 (2019).
> >
> > [2] Gao, Shang-Hua, et al. "Res2net: A new multi-scale backbone architecture." IEEE transactions on pattern analysis and machine intelligence 43.2 (2019): 652-662.
> >
> > [3] Liu, Ze, et al. "Swin transformer: Hierarchical vision transformer using shifted windows." Proceedings of the IEEE/CVF International Conference on Computer Vision. 2021.

---

> > > ### Comment · Reviewer_pMpS · 2022-12-11
> > > **Follow-up questions**
> > >
> > > Could the proposed model be used in generative models, such as [5][6] that contains a bottom-up energy-based model, since the experiments are conducted mainly on discriminative models? If not, I would like to see some discussions as this is an important limitation. If so, is there any change needs to be made? I think it could make this work more solid if the author could add some discussions about this.
> > >
> > > [5] "Representation learning: A statistical perspective." Annual Review of Statistics and Its Application 7.1 (2020): 303-335.
> > > [6] "A theory of generative convnet." International Conference on Machine Learning. PMLR, 2016.

---

### Official Review · Reviewer_mVad · 2022-10-24

**Confidence:** 3
**Correctness:** 3
**Technical Novelty And Significance:** 3
**Empirical Novelty And Significance:** 3
**Recommendation:** 6

**Clarity, Quality, Novelty And Reproducibility:**

There are several spelling mistakes. Just to point out a few:
Can compresses  can compress
Hebbien  Hebbian
Running a grammar/spell checker would be useful.

In general, the paper is very well written. I pointed out above a couple of items that would benefit from further clarification.


**Strength And Weaknesses:**

Strengths

The paper introduces a fresh look at how to compare different architectures and their associated representations of features.

The proposed metrics claim to take into account the model complexity and data efficiency, which are either not incorporated or only indirectly addressed in other metrics such as when merely comparing task performance.

The paper presents an extensive empirical demonstration of the use of the proposed metrics across multiple tasks, datasets, and interesting problems such as model scaling.

Weaknesses

The main results are shown in somewhat unorthodox ways. The rank variable used throughout the figures is a bit hard to interpret. It would be great to expand on the second paragraph of the Experiments section to better explain how this is computed, given that this is critical to interpreting all the figures. Further, it would also be useful to understand how these values relate to other metrics such as accuracy in the datasets, performance for a given number of training examples, and even to show the MDL scores themselves.

What is less clear to me about the model switching proposed here is who and when decides when to switch models. What would be particularly useful is a strategy to be able to switch across models for an arbitrary novel task and dataset. It is not clear to me that this is what the authors are showing. Given the results, it is less interesting to decide a posteriori, which models performed best and when. Perhaps the authors are showing predictability in model switching to new tasks/data but this was not clear to me.

I love theorems and theorems can obviously be more valuable than empirical measures. However, it was a bit difficult to follow in this case, what the connection was between Theorem 1 and the empirical results shown in the subsequent figures (also given that some of the model assumptions are not followed in the architectures as the authors clearly spell out).


**Summary Of The Paper:**

The authors introduce a metric to compare representations in deep learning architectures. This metric is based on thinking of representations as a model selection problem and using the minimum description length principle. The authors also propose the intriguing idea of model switching where different architectures can be used depending on the task demands and characteristics. The metric is applied to traditional architectures for vision encoders across multiple tasks.

**Summary Of The Review:**

This is an interesting and fresh look at the problem of how to characterize and compare different models and their representations. An ideal metric for comparison would take into account not just how the models perform in one or a handful of tasks, but rather a more comprehensive view of the complexity of the architecture, their sample efficiency in training, performance, and generalization. This work is a step in the right direction by introducing new metrics that can satisfy many of these requirements.

---

> ### Author Response · Authors · 2022-11-14
> **Author's response**
>
> Thank you for the review and taking the time to understand and consider our work.
>
> > It would be great to expand on the second paragraph of the Experiments section to better explain how this is computed, given that this is critical to interpreting all the figures.
>
> We will explain more clearly how the average ranking is computed in the experimental section.
>
> > Further, it would also be useful to understand how these values relate to other metrics such as accuracy in the datasets, performance for a given number of training examples, and even to show the MDL scores themselves.
>
> Please see the common response where we present a comparison using average accuracy v.s. average ranking on multiple datasets. The MDL score for all datasets and representations are presented in Table 7 in Appendix.
>
> > What is less clear to me about the model switching proposed here is who and when decides when to switch models.
>
> Once the prior switching strategy is defined, p(\xi_t|D_<t) can be computed and updated in an online fashion. That is the algorithm itself decides how the readout models should be used for the next step prediction based on their performance on previously observed data. The regret bounds guarantee the codelength to be not far away from the best model.
>
> > What would be particularly useful is a strategy to be able to switch across models for an arbitrary novel task and dataset.
>
> Theoretically, the datapoint could come from a novel task or dataset, since prequential MDL and model switching should work for arbitrary sequences including non-stationary ones. In our work, however, we don't showcase on novel task because representation learning benchmarks, such as VTAB, are i.i.d datasets. We reckon non-stationary data sequence is an interesting area for future work.
>
> > It was a bit difficult to follow in this case, what the connection was between Theorem 1 and the empirical results shown in the subsequent figures (also given that some of the model assumptions are not followed in the architectures as the authors clearly spell out).
>
> Please see the regret bounds section in the common response. We are going to improve the explaination in the theory section.
>
> > There are several spelling mistakes.
>
> We will do a grammar and typo check on the paper.

---

### Official Review · Reviewer_GYkn · 2022-11-02

**Confidence:** 4
**Correctness:** 4
**Technical Novelty And Significance:** 2
**Empirical Novelty And Significance:** 2
**Recommendation:** 6

**Clarity, Quality, Novelty And Reproducibility:**

For details, please see strengths & weaknesses above.

I think clarity can be improved (see "Organisation and state of the manuscript").

The quality of the content is generally good, but the benefits and novelty of the proposed method are not entirely clear to me (see "Novelty and contextualisation", "Benefits and evaluation of the proposed approach", and "Discussion of limitations").

With some additional details, the method should be reproducible. I would urge the authors to additionally release their code.


**Strength And Weaknesses:**

### Strengths
- Given the importance of self- and unsupervised representation learning and the variety of methods to do so, evaluating the learnt representations fairly and thoroughly is of great interest to the community. With this submission, the authors provide an interesting approach for evaluating the model representations beyond linear probing.
- The authors evaluate the proposed method on a wide range of experiments, which allows for discussing and analysing differences in the learnt representations with respect to different dimensions of interest (pre-training objective, model size, down-stream dataset.)
- The idea of using MDL to evaluate the model representations is convincing, as it allows to do so simultaneously with models of different complexity whilst still capturing the representations quality in a single number.

### Weaknesses
While I find the idea of the proposed method intriguing, I have issues with several aspects of the current manuscript.

- **Novelty and contextualisation:** The method is presented with an emphasis on the idea of using MDL for evaluating the learnt representations, which gives the impression that this is in itself novel. However, as the authors also state in the background section, Voita & Titov (2020), as well as Yogatama et al. (2019) have already investigated MDL for evaluating representations. Unfortunately, from the current discussion of the related work, the differences to prior work and the novel aspects and their relevance of the proposed approach do not become entirely clear. More generally, many statements lack appropriate references (e.g., the entire introduction has only 3 references), which will make it difficult for readers unfamiliar with the topic to follow.
- **Benefits and evaluation of the proposed approach:** While the proposed approach is generally well-motivated, the practical benefits over linear readout models do not become clear. How do the rankings produced by the proposed approach compare to rankings obtained from simple linear readout models? In what sense are the rankings "better"? What are tangible benefits?
- **Discussion of limitations**: One of the key advantages of linear probing or KNN evaluations of the learnt representations is their simplicity; i.e. the result is largely independent of any hyperparameter settings and easy to obtain and compare. The proposed approach combines many different DNN architectures with varying complexity, which all need to be trained and more heavily depend on the optimisation procedure and hyperparamters, which makes the results potentially less stable and requires careful parameter tuning. I would appreciate if the authors could extend the discussion on this.
- **Organisation and state of the manuscript:** In its current form, the manuscript is difficult to read, mainly due to the following reasons.
	- The structure of the manuscript is not easy to follow and the relevance of some sections is unclear. For example, section 3 abruptly switches to regret bounds and introduces Theorem 1, without discussing why this is relevant. In fact, the theorem is introduced and discussed, only to then say that the assumptions of the theorem do not hold in the context of DNNs, i.e. the context of this work. Similarly, an online version for the MDL computation is introduced and discussed, only to then say that this is not used in this work ("To facilitate experimenting with different switching strategies, we use this 2-stage implementation for all our experiments."); this statement conflicts with the description of the experimental details in section 4 ("We use the online learning framework as described in Section 3"). I would appreciate if the authors could clarify why these sections are relevant and if the discussion around these sections were improved.
	- The mathematical notation could be improved. For example, in the fourth line of the "Read-out Model Switching" paragraph, the formula contains a "p_{\xi_t}", which is referred to as "p_k" afterwards. Later, a p(k) is used, which refers to p({\xi_1}) as far as I understand (however, p(k) = 1 iff k=1, which should simplify the L_{BM} equation further), which adds unnecessary notational ambiguity and makes it difficult to follow.
	- Finally, as discussed above, I would highly appreciate if the authors could add citations when appropriate (especially in the introduction).
- **Experimental protocol and ablations**: Some of the experimental details remain unclear. How long are the individual models trained on each time step? Is the prequential method stable w.r.t. the order of the data? Why are some of the datasets reshuffled? This seems to be non-standard and I fail to see why this is required. What happens without reshuffling?

Minor: I would recommend to carefully check the paper for grammatical correctness for the final version.

**Summary Of The Paper:**

The authors of this submission propose to use the Minimum Description Length (MDL) principle to evaluate the quality of the learnt representations in DNNs. In particular, they follow the prequential approach to estimate the description length, which assumes that *good* representations should allow for training prediction models for down-stream tasks in a data-efficient manner.

Concretely, the authors train a set of K models in a prequential manner (with SGD) and use a switching strategy to choose the best model for a given number of training samples. The final code length is then computed as the cumulative prediction loss of this model ensemble.

To evaluate their proposed method, the authors perform experiments with various DNN architectures (ResNets and ViTs of different sizes) on a wide range of datasets. Further, the authors discuss insights obtained from their experiments, such as regarding the pre-training paradigm used to train the encoders (e.g., supervised vs. self-supervised), the model size, or the transfer learning dataset.

**Summary Of The Review:**

As discussed above, I think the proposed approach is interesting and could be a valuable addition to the literature on evaluating DNN representations. However, in its current state (see weaknesses), I think the paper is not ready for publication.

EDIT: After the authors' response, some of my concerns were addressed and I therefore raise my score to "6: marginally above the acceptance threshold".

---

> ### Author Response · Authors · 2022-11-14
> **Author's response (1/2)**
>
> Thank you for the thoughtful review of our work.
>
> > Novelty and contextualisation: Unfortunately, from the current discussion of the related work, the differences to prior work and the novel aspects and their relevance of the proposed approach do not become entirely clear.
>
> Our work differs to previous works that use MDL for evaluation in 3 aspects:
> 1) Previous works use only a single readout protocol, while we demonstrate that our method can be applied for combining different readout protocols regardless of the backbone being frozen or fine-tuned. With the comparison provided in the common response, we hope to convince the reviewer that a change in evaluation protocol leads to inconsistent model selection and this is an issue for proper evaluation of the representations.
> 2) We use an online framework to compute prequential MDL scores which has not been used previously to evaluate representation. It leads to shorter description lengths [1] and crucially enables the model switching between readout-methods at each timestep.
> 3) Our framework gives insights into properties of the representations. As we show in the experimental section, we can inspect the performance and preference of protocol under different dataset sizes. This can guide us in choosing the best way to use the representation.
>
> > More generally, many statements lack appropriate references (e.g., the entire introduction has only 3 references), which will make it difficult for readers unfamiliar with the topic to follow.
>
> We will clarify our contributions, add references to the introduction section, improve the background section and grammar check the full paper.
>
> > Benefits and evaluation of the proposed approach: How do the rankings produced by the proposed approach compare to rankings obtained from simple linear readout models? In what sense are the rankings "better"? What are tangible benefits?
>
> Please see comparisons to other evaluations and discussions in our common response.
>
> > Discussion of limitations: The proposed approach combines many different DNN architectures with varying complexity, which all need to be trained and more heavily depend on the optimisation procedure and hyperparamters, which makes the results potentially less stable and requires careful parameter tuning.
>
> We agree that linear probing and KNN are valuable in that we can get a quick idea of the representation quality while developing algorithms. However, it is a fact that various papers and representation learning methods do use or prefer different readout methods beyond KNN and linear probing. This issue of the evaluation protocol has been hinted at in the representation learning literature and we provide some concrete evidence in our common response. The purpose of our paper is to provide a rigorous method for evaluating representations even when a multitude of readout methods are in use.
>
> The careful tuning is necessary regardless of using MDL or not, since different protocols are used. From our experiment, hyperparameters related to switching are very stable across all representations and, therefore, are fixed for all.
>
> > Section 3 abruptly switches to regret bounds and introduces Theorem 1, without discussing why this is relevant. In fact, the theorem is introduced and discussed, only to then say that the assumptions of the theorem do not hold in the context of DNNs
>
> Please see the common response regarding regret bounds. We will improve the explanation to why it's necessary to discuss in this section.
>
> > An online version for the MDL computation is introduced and discussed, only to then say that this is not used in this work ("To facilitate experimenting with different switching strategies, we use this 2-stage implementation for all our experiments."); this statement conflicts with the description of the experimental details in section 4 ("We use the online learning framework as described in Section 3")
>
> The online learning framework is used for updating parameters of the readout models. In the 2-stage implementation (see Algorithm 2 in Appendix), the first stage uses the same online learning framework. So it is not conflicting with our statement. The single stage and 2-stage implementations are strictly equivalent. We include 2-stage implementation because we would like to provide more details for our experimental setup. But if this is confusing and deemed unnecessary, we can remove the 2-stage implementation.
>
>
> [1] Sequential Learning Of Neural Networks for Prequential MDL. arXiv preprint arXiv:2210.07931 (2022).

---

> > ### Author Response · Authors · 2022-11-14
> > **Author's response (2/2)**
> >
> > > The mathematical notation could be improved. For example, in the fourth line of the "Read-out Model Switching" paragraph, the formula contains a "p_{\xi_t}", which is referred to as "p_k" afterwards. Later, a p(k) is used, which refers to p({\xi_1}) as far as I understand (however, p(k) = 1 iff k=1, which should simplify the L_{BM} equation further), which adds unnecessary notational ambiguity and makes it difficult to follow.
> >
> > \xi_{1:T} are random variables which take values in [1,..,K] and these are identities of the readout models (note that the sample spaces are the same for all the \xi_t). p_k(.) is the predictive distribution of the kth readout model. The BM prior can be seen as having an initial probability p(\xi_1=k), and the transition probabilities are p(\xi_t|\xi_t-1)=1 if \xi_t=\xi_t-1; 0 otherwise. We use p(k) as a simplified notation for p(\xi_1=k). We will clarify this and revert to p(\xi_1=k) in the final version of the paper.
> >
> > > I would highly appreciate if the authors could add citations when appropriate (especially in the introduction)
> >
> > We will add more references and citations, especially to the introduction section.
> >
> > > How long are the individual models trained on each time step?
> >
> > Table 3 in Appendix summarizes the hyperparameters. Specifically, the number of replay streams is equivalent to the training steps on each time step.
> >
> > > Is the prequential method stable w.r.t. the order of the data?
> >
> > Please see the common response for the ordering of the data.
> >
> > > Why are some of the datasets reshuffled? This seems to be non-standard and I fail to see why this is required. What happens without reshuffling?
> >
> > We decided to reshuffle the datasets because we discovered that there are few datasets in VTAB that are not i.i.d and have a shift between train and test subsets. Please see Appendix A.2 for details. We think the VTAB benchmark should address this problem, since the shift makes results on the benchmark less reliable, i.e. it’s not evaluating what we think it is. We will publish our version of the datasets and the code used for our evaluations.

---

> > > ### Comment · Reviewer_GYkn · 2022-11-15
> > > **Thank you for the detailed rebuttal**
> > >
> > > I thank the authors for their detailed response and for their effort put into addressing my concerns. I still have the following open questions / remarks.
> > >
> > > First, the paper would significantly benefit from verifying that the model switching indeed works as intended. I.e., for a given dataset size (e.g., the full dataset), it would be very helpful to see that (1) the "chosen" model via the switching strategy indeed performs best if trained from scratch on the respective data and (2) the resulting ranking between different models is consistent with the one obtained via the proposed protocol. This would more easily convince the reader that the ranking is indeed reliable and is not (significantly) affected by the iterative training protocol as used in the method (i.e. training on each time step with different amounts of data).
> > >
> > > Second, while most of my concerns regarding the *method* were adequately addressed, I am still concerned about the *clarity* of the final paper, which is difficult to judge as of now, as it seems that major changes to the manuscript will be necessary.
> > >
> > > Third, regarding the re-shuffled datasets, I would still urge the authors to also show the results for the non-shuffled datasets, at least in the supplement. There is no principled reason why the evaluation protocol should not work in this case.
> > >
> > > In summary, taking the answers into account, I believe the proposed method to be a valuable contribution to the literature and therefore raise my score to 6, but some hesitations regarding the clarity of the manuscript remain.

---

> > > > ### Author Response · Authors · 2022-11-18
> > > > **Author's reply**
> > > >
> > > > Thank you for the suggestions. We believe they have helped us to improve the manuscript!
> > > >
> > > > First, we would like to confirm that model switching results in a shorter codelength compared to the codelength of any individual readout model on all datasets investigated. Thus, the model switching indeed works. However, we reckon that there can be differences between online and offline training. Therefore, we don’t expect them to always agree perfectly at each dataset size. Nevertheless, we believe the online learning framework used provides a reasonable and feasible approximation to our metric and, hence, is a step forward in evaluating representations. We hope future works can further improve online learning.
> > > >
> > > > We have updated the manuscript. We would appreciate it if the reviewer could have another look.
> > > >
> > > > Unfortunately, we are not able to report results on original VTAB datasets within the deadline of the rebuttal, as this requires rerunning all the experiments. We can add the results in the Appendix when they become available. However, we caution against a direct comparison of our MDL result with ERM in case of non i.i.d data and distribution shift, because: 1) ERM relies on the assumption of i.i.d-ness; 2) in a non-stationary setting, MDL prefers representation that can adapt to the changes in distribution fast.

---

### Official Review · Reviewer_Jszx · 2022-11-03

**Confidence:** 3
**Correctness:** 3
**Technical Novelty And Significance:** 3
**Empirical Novelty And Significance:** 2
**Recommendation:** 6

**Clarity, Quality, Novelty And Reproducibility:**

I could understand the contents of the paper, but their were some grammatically incorrect sentences that needs to be fixed for the clarity. The claimed work seems original. Algorithms, architectures, and hyper-parameters are well documented for reproducibility.

### Typos

Section 3, first paragraph, line 7: "a fixed model"

Caption for Figure 5: "Data efficiency"

**Strength And Weaknesses:**

## Strength

* **Novel approach in evaluating representation quality.** Authors provide a new way of assessing representation quality, and their algorithm for obtaining this metric is straightforward.

* **Adopting multiple readout model.** Good representations may not store the crucial information of data in a linear shape. Adopting multiple readout model can alleviate this concern. It was interesting to see data efficiency of each readout models.

## Weakness

* **Justification on using model switching process.** The switching model forces to change the status of $\eta$ in after some time span, but the case shown in Figure 1, for example, the optimal selection should stick to one model after observing enough amount of data. The switching process is not likely to achieves the optimal sequence of readout model selection. This gap must be discussed. Instead of following a process with discrete status, using an ensemble method may relieve this problem, vaguely speaking.

* **Justification on regret bound.** As acknowledged in the paper, the regret bound (2) holds for exponential family with MLE $\theta$. Thus, this upper bound crucial for constructing Algorithm 1 may not hold when the readout models include MLP.

* **The claimed metric needs an empirical evidence about the quality of the metric** to assert that the proposed method is feasible. E.g., comparing the MDL rank with quantitative performance metric of the represented data on practical tasks such as classification or domain adaptation.

**Summary Of The Paper:**

Authors claim a new metric on the quality of representation, Minimum Description Length (MDL) with model switching.
With the representation model and readout model given, the proposed metric is the optimized upper bounds for the regret in switching readout model with respect to MDL loss.
The proposed metric can also evaluate data efficiency on online learning.
For the experiment, authors use ImageNet and VTAB database with ResNet-50 and ViT/B16 architectures and rank the representation quality with the claimed metric.

**Summary Of The Review:**

Although the authors provides a technical algorithm for obtaining a novel representation quality metric, it lacks theoretical justification and empirical evidence that the metric is feasible to assess quality of representation.
Thus, I reject this submission.

**Post Rebuttal**

The revised script included discussions and materials that I expected for supporting the argument of the paper. Thus, I have updated my scores.

---

> ### Author Response · Authors · 2022-11-14
> **Author's response**
>
> Thank you for the review. Below we address the questions and concerns raised:
>
> > The switching model forces to change the status in after some time span, but the case shown in Figure 1, for example, the optimal selection should stick to one model after observing enough amount of data. The switching process is not likely to achieves the optimal sequence of readout model selection.
>
> The reviewer misunderstood how the model switching works. The model is not forced to change. The switching strategy is a prior probability. The model can choose to stick with a member of the readout models if there is sufficient evidence suggesting a particular readout model is better at compressing the data. In Appendix A.7, we plot the posterior of the readout models and we can clearly see that the model is not forced to change. Whether the switching achieves the optimal is exactly what the regret bound is answering.
>
> > As acknowledged in the paper, the regret bound (2) holds for exponential family with MLE . Thus, this upper bound crucial for constructing Algorithm 1 may not hold when the readout models include MLP.
>
> Please see the common response on regret bounds. The regret bound is important to discuss, however Algorithm 1 works as long as it converges asymptotically to the optimal solution.
>
> > The claimed metric needs an empirical evidence about the quality of the metric to assert that the proposed method is feasible
>
> Please see the common response where we provide comparisons to other evaluation methods on VTAB datasets.
>
> > I could understand the contents of the paper, but their were some grammatically incorrect sentences that needs to be fixed for the clarity.
>
> We will do grammar checks and fix all the typos.
>
> We hope we have clarified what model switching achieves in the above - if so, we would be grateful if you could take another look. If any questions remain, please let us know and we will do our best to answer them.

---

### Author Response · Authors · 2022-11-14
**Common response for all reviewers**

We thank all reviewers for their thoughtful comments and valuable suggestions. We address the shared concerns below:

# Comparisons with Linear and MLP evaluations
We provide comparisons the reviewers suggested. For both linear and MLP evaluations, we perform a hyperparameter search on learning rate, weight decay and EMA with a batch size 1024. We split the dataset into training and validation where the validation set contains 10% of the data. We choose the best performing hyperparameter on validation and report top1 accuracy of the test set. We note that using a combination of MLPs is **NOT** a standard practice and there is no consensus on how to combine results from different probes and why one is preferable than another. In the comparison provided, we use the highest accuracy among the readout models.

## Pre-training Losses Experiment
We summarize the model selection results for pre-training losses according to 1) linear evaluation with average accuracy; 2) linear evaluation with average ranking; 3) combinations of MLPs with average raking; 4) our method.

1). Linear evaluation with average accuracy on VTAB:
|Model	|		Avg Top1 Acc (%)|
| :- | :-: |
|ResNet50-BYOL          	|77.93|
|ViT/B16-DINO           	|	76.01|
|ResNet50-SUP		|75.51|
|ResNet50-SimCLR        	|74.35|
|ViT/B16-MAE            	|	74.10|
|ViT/B16-SUP     	|	72.46|

2). Linear evaluation with average ranking on VTAB:
|Model|				Avg Rank|
| :- | :-: |
|ViT/B16-DINO | 2.26|
|ResNet50-BYOL | 2.37|
|ResNet50-SUP | 3.82|
|ViT/B16-MAE | 3.84|
|ViT/B16-SUP | 4.18|
|ResNet50-SimCLR | 4.53|

3). Combinations of MLPs with average raking:
|Model|Avg Rank|
| :- | :-: |
|ViT/B16-DINO | 1.84|
|ResNet50-BYOL | 2.63|
|ViT/B16-MAE | 3.24|
|ViT/B16-SUP | 4.21|
|ResNet50-SUP | 4.45|
|ResNet50-SimCLR | 4.63|

4). Our method:
|Model|Avg Rank|
| :- | :-: |
|ViT/B16-DINO | 1.89|
|ResNet50-BYOL | 2.53 |
|ViT/B16- MAE | 3.79|
|ViT/B16-SUP	| 4.16 |
|ResNet50-SimCLR | 4.21|
|ResNet50-SUP | 4.42 |

First, we note the two summarization methods (avg accuracy and avg rankings) don’t have the same results for model selection. The reason why we should not averaging over multiple datasets is discussed in [1]. In brief, “If the results on different data sets are not comparable, their averages are meaningless” and “averages are also susceptible to outliers''. Since the VTAB benchmark consists of 19 datasets for classification tasks of very different nature, averaging over the accuracy is an ill-conceived practice. Therefore, we omit the averaging accuracy over datasets in the comparison below.

## Scaling Experiment
We summarize the model selection results for model scaling according to 1) linear evaluation with average ranking; 2) combinations of MLPs with average raking; 3) our method.

1). Linear evaluation:
|Model|Avg Rank|
| :- | :-: |
|ResNet101-BYOL | 2.84|
|ResNet152-BYOL | 3.11|
|ViT/L16-MAE | 3.42|
|ResNet50-BYOL | 4.24|
|ResNet50-SUP | 6.89|
|ViT/B16-MAE	| 6.89|
|ResNet101-SUP | 6.95|
|ViT/L16-SUP	| 7.05|
|ViT/B16-SUP	| 7.34|
|ResNet152-SUP | 7.47|
|ViT/S16-SUP	| 10.26|
|ViT/S16-MAE | 11.53|

2). Combinations of MLPs:
|Model|Avg Rank|
| :- | :-: |
|ViT/L16-MAE | 2.53|
|ResNet152-BYOL | 3.71|
|ResNet101-BYOL | 3.92|
|ResNet50-BYOL    | 4.26|
|ViT/B16-MAE          | 5.34|
|ViT/B16-SUP       	| 6.47|
|ResNet152-SUP     | 7.00|
|ResNet101-SUP     | 7.47|
|ResNet50-SUP      	| 7.55|
|ViT/L16-SUP       	| 8.50|
|ViT/S16-SUP      	| 10.61|
|ViT/S16-MAE          | 10.63|

3). Our method:
|Model|Avg Rank|
| :- | :-: |
|ResNet101-BYOL |		3.05|
|ResNet152-BYOL	|	3.26|
|ViT/L16-MAE		|	3.74|
|ResNet50-BYOL	|	4.05|
|ViT/B16-SUP		|	6.00|
|ViT/B16-MAE		|	6.21|
|ResNet50-SUP	|	7.11|
|ResNet101-SUP	|	7.21|
|ResNet152-SUP	|	7.47|
|ViT/L16-SUP		|	9.11|
|ViT/S16-SUP		|	10.37|
|ViT/S16-MAE		|	10.42|

We observe that allowing multiple readout protocols has a significant impact on model selection. Other works have observed this as well [2,3]. However, we are not aware of any previous work combining results under different protocols to provide a model selection criteria.
Our method solves this issue by allowing any probing protocols to be used and taking into account their complexity. And for prequential MDL, model complexity and the speed of learning are two sides of the same coin. Therefore, with our evaluation framework, performance in the small data regime is an important factor and representations performing worse with little data are penalized in the final score.
We would also like to emphasize that our evaluation framework provides important insights into the properties of the representations, which is not readily available with the linear evaluation.
We will add these comparisons and discussions to the paper.

---

> ### Author Response · Authors · 2022-11-14
> **Common response for all reviewers (continued)**
>
> # Regret bounds
> The regret bound guarantees the upper bound of the gap between prequential MDL codelength and the shortest codelength achieved by the optimal model in hindsight. A sublinear regret, such as O(log N) achieved by fixed share, means that the algorithm gets to the optimal relatively fast. Therefore, it is an important aspect to discuss in our opinion. While we acknowledge that some assumptions do not hold in practice, this is quite common for complicated problems/algorithms, especially for Neural Networks. E.g.: We can prove the convergence rate of SGD for convex problems but not for Neural Networks (NN). In a similar way, we are not able to prove the exact regret bound with NN yet, but with an efficient training method, it should converge asymptotically to the optimal solution. To develop efficient online training methods and relax the assumptions for the regret bounds are areas for future research. We will explain more about the regret bounds in the paper.
>
> # Data Ordering
> Theoretically, different permutations of the data sequence results in different codelengths. We reckon that this is an area for future work. [4] suggests taking the permutation with the shortest codelength, but it is computationally too expensive.  [5] notes that the codelength is invariant over orderings when a Bayesian predictor is used in prequential MDL.  [6] points out that while the ordering of data affects the codelength, it  does not appear to cause too much trouble in practice. Here, we provide an empirical evaluation of the variance of our metric. We evaluate ResNet50-SUP representation on Cifar100 with 5 permutations and obtain the MDL score of 55906 with a standard deviation of 134. To give an idea, the representations that are ranked immediately above and below achieve the MDL score of 50969 and 60549 respectively. Therefore, we believe our evaluation method is stable and reliable. We would also like to point out that the current practice of comparing the test accuracy is not immune to randomness, e.g. initialization, dataset split and randomization during training. We are going to add data ordering into the discussion section of the paper and provide the empirical results in the Appendix.
>
> [1] Janez Demsar. Statistical comparisons of classifiers over multiple data sets.The Journal of Machine learning research, 7:1–30, 2006.
>
> [2] Cinjon Resnick, Zeping Zhan, and Joan Bruna. Probing the state of the art: A critical look at visual representation evaluation. CoRR, abs/1912.00215, 2019.
>
> [3] Kaiming He, Xinlei Chen, Saining Xie, Yanghao Li, Piotr Doll ́ar, and Ross B. Girshick. Masked autoencoders are scalable vision learners.CoRR, abs/2111.06377, 2021.
>
> [4] Rissanen, Jorma. "Stochastic complexity and modeling." The annals of statistics (1986): 1080-1100.
>
> [5] Hansen, Mark H., and Bin Yu. "Model selection and the principle of minimum description length." Journal of the American Statistical Association 96.454 (2001): 746-774.
>
> [6] De Rooij, Steven, and Peter D. Grünwald. "Luckiness and regret in minimum description length inference." Philosophy of Statistics. North-Holland, 2011. 865-900.

---

### Author Response · Authors · 2022-11-18
**New Revision**

Dear reviewers,

Thank you all for the valuable feedback. We uploaded a new revision with various improvements and addressing the concerns. We would appreciate if you could take another look!

---

### Decision · Program_Chairs · 2023-01-20

**Decision:**

Accept: poster

**Justification For Why Not Higher Score:**

This is just a borderline accept paper.

**Justification For Why Not Lower Score:**

No reviewers want to reject the paper.

**Metareview: Summary, Strengths And Weaknesses:**

This paper studies the evaluation problem in representation learning. The authors treat the evaluation of representations as a model selection problem and propose to use the Minimum Description Length (MDL) principle to devise an evaluation metric.  The authors demonstrate the effectiveness of the proposed metric by presenting numerical results for pre-trained vision encoders with different network architectures, including ResNet and ViT, and various objective functions, including supervised and self-supervised objectives, on a bunch of downstream tasks.  The metric is also compared with accuracy-based approaches to show its superiority. The paper receives a total of 5 reviews, including one (Reviewer 4XSa) with very low confidence that the AC has to downweight when making the final decision. All reviewers agree that the proposed method in the paper is interesting, novel, and has extensive empirical demonstration. Concerns and questions, such as incomplete reference, lack of discussion about limitation, unclear experimental details and presentation/organization issue, have been addressed and replied by the authors during the rebuttal. All 5 reviewers feel positive on the paper after the rebuttal, and lean to accept it (marginally above the acceptance bar). The AC has carefully read the comments, rebuttal, revised paper, and had an internal meeting with the reviewers. The AC agree with the reviewers’ opinion, thus recommending accepting the paper as a poster.

**Note From Pc:**

if the above contains the word "oral" or "spotlight" please see: "oral" presentation means -> notable-top-5% and "spotlight" means -> notable-top-25%. As stated in our emails, we are disassociating presentation type from AC recommendations

**Summary Of Ac-Reviewer Meeting:**

The AC had a discussion with reviewers. Reviewer GYkn shares hie/her opinion: The authors present a sufficiently novel methodology for evaluating model representations with a thorough and extensive experimental evaluation, which meets the bar for a publication at the conference, but I am unsure about the clarity of the current version of the submission. That said, I am currently leaning towards recommending the paper for acceptance. Even though not all the reviewers joined the discussion, their comments in the reviews and feedbacks to the rebuttal have confirmed that the current paper is OK to be accepted. The AC finally recommends accepting the paper as a poster.